# Non-Clairvoyant Scheduling with Progress Bars

**Ziyad Benomar**
CREST, ENSAE, Ecole Polytechnique,
Fairplay joint team,
Palaiseau, France
`ziyad.benomar@ensae.fr`

**Romain Cosson**
Courant Institute of Mathematical Sciences
New York University
`rc6142@nyu.edu`

**Alexander Lindermayr**
Simons Institute for the Theory of Computing
UC Berkeley
`alex.lindermayr@berkeley.edu`

**Jens Schlöter**
Centrum Wiskunde & Informatica (CWI)
Amsterdam, The Netherlands
`Jens.Schloter@cwi.nl`

## Abstract

In non-clairvoyant scheduling, the goal is to minimize the total job completion time without prior knowledge of individual job processing times. This classical online optimization problem has recently gained attention through the framework of learning-augmented algorithms. We introduce a natural setting in which the scheduler receives continuous feedback in the form of progress bars—estimates of the fraction of each job completed over time. We design new algorithms for both adversarial and stochastic progress bars and prove strong competitive bounds. Our results in the adversarial case surprisingly induce improved guarantees for learning-augmented scheduling with job size predictions. We also introduce a general method for combining scheduling algorithms, yielding further insights in scheduling with predictions. Finally, we propose a stochastic model of progress bars as a more optimistic alternative to conventional worst-case models, and present an asymptotically optimal scheduling algorithm in this setting.

## 1 Introduction

We study the fundamental problem of scheduling on a single machine: there are $n$ jobs, each with a processing time $p_j$, available at time 0, and the goal is to preemptively (i.e., jobs can be interrupted and resumed later) schedule them to minimize the sum of the completion times. In the *clairvoyant* (or *offline*) setting, i.e., when all processing times are known to the scheduler, the optimal solution is to schedule jobs in increasing order of processing times. This rule is known as Shortest Processing Time First, or as Smith's rule in reference to a 1956 paper that first formalized it [Smi56]. In many practical applications, such as operating systems or high-performance computing, the scheduler does not know the jobs' processing times upfront. This restriction led to the model of *non-clairvoyant* (or *online*) scheduling, where the scheduler learns about the processing time of a job only when this job is completed. For this online problem, introduced in the early 90s [FST94, SWW91, MPT94], a classical algorithm is Round-Robin, which simply allocates the same amount of computing power to all unfinished jobs. It approximates the optimal solution within a factor (called competitive ratio) of 2, and it is well-known that this value is optimal among non-clairvoyant algorithms [MPT94].

The clairvoyant and non-clairvoyant models have a stark contrast in their *information model*, which is the amount of information (feedback) the online algorithm receives over time. In this paper, we introduce and study a model of scheduling with *progress bars* that interpolates between these two settings. Non-clairvoyant scheduling has already been investigated through the lens of *learning-augmented algorithms* (aka *algorithms with predictions*) [LV21, MV20, LM25b]. Therein, algorithms

39th Conference on Neural Information Processing Systems (NeurIPS 2025).

are equipped with additional, possibly erroneous information (predictions) to overcome limitations of the zero knowledge assumption and hence improve over classical lower bounds. The goal is to establish strong performance guarantees for perfect predictions (*consistency*) that degrade gracefully as the quality of the predictions worsens (*smoothness*), and which stay bounded for arbitrarily bad predictions (*robustness*). This line of research studied various prediction models for non-clairvoyant scheduling, such as (partial) predictions on the unknown processing times (input predictions) [PSK18, WZ20, ALT21, ALT22, IKQP23, BP23, BP24], or predictions on the optimal scheduling order (action predictions) [DIL⁺22, EKMM24, LM25a, LLMS23]. A shortcoming of these models is that they give predictions *a priori*, hence relying only on static information such as job identifiers or descriptions. This limits their applicability in natural situations where information might initially be scarce or extremely unreliable, but where its amount and reliability increases as the scheduler interacts with the system.

Given these limitations, we ask the following question: *Is there an information model that allows algorithms to refine their decisions over time?* Our main motivation for this question is estimation techniques such as profiling [HRB18, WBB⁺17, XGWC23], which *learn* about jobs as they are being executed: while initially we have no good estimate of processing times, we can expect to give improved predictions after some processing. Inspired by this, we propose a formal model based on *progress bars*: while processing a job, we receive an estimated indication of how much percentage of the job has been completed, revealed at different levels of *granularity* (e.g., frequency, time, or levels of updates). Clearly, we cannot expect that the progress bar is perfectly accurate, motivating the design of scheduling algorithms for adversarial and stochastic progress bars.

## 1.1 Our results

A *progress bar* for a job $j \in [n] := \{1, \ldots, n\}$ is formally defined as a function $\varphi_j : [0, 1] \to [0, 1]$, that is non-decreasing and satisfies $\varphi(0) = 0$ and $\varphi(1) = 1$. For all $x \in [0, 1]$, $\varphi_j(x)$ represents the *displayed* progress when job $j$ has an *actual* progress of $x$, that is, it has been executed for a fraction $x$ of its total processing time. The value $\varphi_j(x)$ can be interpreted as an estimate of $x$ at any time during processing. In particular, we can model clairvoyant scheduling as having accurate progress bars $\varphi_j : x \mapsto x$ for all $j$ (cf. Figure 1a),[1] while the non-clairvoyant setting corresponds to uninformative progress bars $\varphi_j : x \mapsto \mathbb{1}(x = 1)$ (cf. Figure 1b). We say that the progress bar $\varphi$ has a *granularity* $g \in \mathbb{N}$ if it is a step function taking $g$ values besides 0 and 1.

From the scheduler's point of view, after allocating $e_j \leq p_j$ units of computing to job $j$, the displayed progress of job $j$, denoted by $X_j(e_j) = \varphi_j(e_j/p_j)$, provides an estimate of the actual progress of job $j$, equal to $x_j = e_j/p_j$. It can also be interpreted as a prediction $\pi_j = e_j/X_j(e_j)$ on the job length, which usually refines as more computing is allocated to that job.

The problem of scheduling with progress bars has a flavor similar to decision with bandit feedback [LS20, BC12]. Indeed, the scheduler faces an explore-exploit tradeoff, where it is incentivized to prioritize jobs for which the progress bar seems to progress faster (*exploit*) but must also allocate some computation to other jobs to refine estimates of their processing times (*explore*). Like in the bandits literature, the study of scheduling algorithms with progress bars lends itself to both an adversarial analysis and to a stochastic analysis. Specifically, we study three models of progress bars, which we describe in detail below.

### 1.1.1 Single signal progress bar

We first consider progress bars of granularity 1. In this case, the progress bar stays at 0 until a fraction $\beta_j$ of job $j$ has been processed, at which point it jumps to a displayed progress of $\alpha$ (where $\beta_j$ may differ from $\alpha$ due to inaccuracies). Finally, it jumps to 1 upon completion of the job. Formally,

$$\varphi_j(x) = \alpha \cdot \mathbb{1}(\beta_j \leq x < 1) + \mathbb{1}(x = 1).$$

**Accurate signal.** We start by investigating the case where $\beta_j = \alpha$ for all jobs $j$. This is equivalent to each job emitting a signal exactly after it has been processed for an $\alpha$ fraction. This setting interpolates between the *non-clairvoyant* ($\alpha = 1$) and *clairvoyant* ($\alpha = 0$) regimes. This setting has

---

[1]There is a minor technical difference: while we learn in the clairvoyant setting about $p_j$ at $j$'s arrival, we need to process $j$ for an $\varepsilon > 0$ fraction to deduce $p_j$ via the accurate progress bar $\varphi_j : x \mapsto x$.

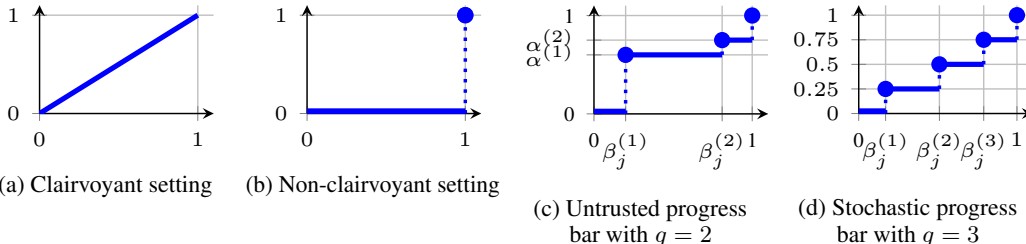

Figure 1: Visualization of different progress bars. The horizontal axis represents the actual progress $x$ of a job and the vertical axis represents the displayed progress $\varphi(x)$.

been studied before in a different context [YT17, GKL$^+$25]. We show in Theorem 2.1 that a simple deterministic algorithm is $(1 + \alpha)$-competitive. This result appears concurrently also in [GKL$^+$25].

**Consistent, robust and smooth algorithm.**   Next, we assume that the signal emission time can be inaccurate, i.e., $\beta_j \neq \alpha$. We demonstrate in Theorem 2.4 that for all $\rho \in (0, 1)$, there is a deterministic algorithm that is $(1 + \alpha)$-consistent, $(1 + 1/\rho\alpha)$-robust, and smooth. The parameter $\rho$ tunes a tradeoff between robustness and smoothness in this algorithm. We also demonstrate in Appendix E that the consistency and robustness bounds can be extended to the more general setting of having $m$ parallel identical machines.

**Improved consistency-robustness tradeoff.**   In Section 2.3, we discuss how our technique yields an algorithm for scheduling with job length predictions that improves over the best consistency-robustness tradeoff in prior work, achieved by time sharing [PSK18]. Interestingly, our algorithm is the first to actually use the numerical values of the predicted job lengths for robustification, while time sharing is a black box technique, which also applies to permutation predictions [LM25a].

### 1.1.2   Untrusted progress bar

We then consider the more general model where the progress bar has *granularity* $g \in \mathbb{N}$, displaying $g$ intermediate progress levels $\alpha^{(1)} < \ldots < \alpha^{(g)}$. We denote for convenience $\alpha^{(0)} = 0$ and $\alpha^{(g+1)} = \beta_j^{(g+1)} = 1$. For each $h \in [g]$, the displayed progress increases from $\alpha^{(h-1)}$ to $\alpha^{(h)}$ when the actual progress reaches $\beta_j^{(h)}$ (cf. Figure 1c for an example). Hence, we formally define for all $x \in [0, 1]$

$$\varphi_j(x) = \sum_{h=1}^{g+1} (\alpha^{(h)} - \alpha^{(h-1)}) \cdot \mathbb{1}(x \geq \beta_j^{(h)}) \,.$$

**General combining algorithm.**   To address this setting, we study a more general problem: how to combine multiple scheduling algorithms. [EKMM24] introduced a randomized strategy to combine multiple permutation predictions, achieving performance close to that of the best prediction, up to a regret term. In Theorem 3.1, we extend and improve their approach to combine arbitrary scheduling algorithms under mild assumptions, even if the algorithms rely on different types of predictions, or none at all. In particular, this combining strategy can be applied in our setting by interpreting a progress bar with granularity $g$ as a collection of $g$ progress bars with granularity 1 each, and combining different instantiations of the algorithm for the single-signal setting.

**Implications in learning-augmented scheduling.**   With our strategy, we can combine Round-Robin with any 1-consistent algorithm in any prediction model, giving a consistency of $1 + o(1)$ and a robustness of $2 + o(1)$ for a broad class of scheduling instances, where the maximum job processing time is small enough compared to the optimal objective value. For this class of instances, our result significantly improves upon previously known consistency-robustness tradeoffs for non-clairvoyant scheduling with predictions, and substantially narrows the gap between the prior state-of-the-art algorithms and the lower bound on the consistency-robustness tradeoff established in [WZ20].

### 1.1.3 Stochastic progress bar

Finally, we study a stochastic model of progress bars with *granularity* $g \in \mathbb{N}$. Specifically, the bar can display $g + 2$ progress levels $(h/g+1)_{0 \le h \le g+1}$, and jumps between these levels according to a Poisson point process with rate $g$. Let $\tilde{\beta}_j^{(1)} \le \ldots \le \tilde{\beta}_j^{(g)}$ denote the first $g$ points of this process, $\beta_j^{(h)} = \min(1, \tilde{\beta}_j^{(h)})$ for all $h \in [g]$, and $\beta_j^{(g+1)} = 1$. An example is given in Figure 1d. Formally,

$$\varphi_j(x) = \frac{1}{g+1} \sum_{h=1}^{g+1} \mathbb{1}(x \ge \beta_j^{(h)}) .$$

Our algorithm follows a repeated explore-then-commit approach: it runs Round-Robin until the progress bar of a job reaches a certain threshold of order $\Theta(g^{2/3})$, then it commits to that job until completion, and then it resumes the exploration phase with Round-Robin on the remaining jobs. We show in Theorem 4.2 that the competitive ratio of this algorithm is $1 + \mathcal{O}(g^{-1/3})$, and we prove in Theorem 4.3 a matching lower bound of $1 + \Omega(g^{-1/3})$ on the competitive ratio of any algorithm.

## 1.2 Notation and organization

There are $n$ jobs $j \in [n]$ with processing times $p_j$ such that $p_1 \le \ldots \le p_n$. We consider scheduling with arbitrary preemptions, so a scheduler can process a job for an infinitesimal small amount of time. The optimal objective value is given by $\text{OPT} = \sum_{i=1}^{n}(n - i + 1)p_i$ [Smi56]. The processing that job $j$ has received until time $t$, also called the elapsed time of job $j$, is denoted by $e_j(t)$. The completion time $C_j$ of job $j$ is the earliest time $t$ that satisfies $e_j(t) \ge p_j$. The total completion time of a scheduling algorithm is equal to $\text{ALG} = \sum_{j=1}^{n} C_j$. For two jobs $i, j \in [n]$, we denote by $d(i, j)$ the delay that $i$ incurs to $j$, i.e., $d(i, j) = e_i(C_j)$. We have (see, e.g., [MPT94]):

$$\text{ALG} = \sum_{j=1}^{n} p_j + \sum_{i=1}^{n} \sum_{j=1}^{i-1} d(i, j) + d(j, i) . \tag{1}$$

We present our models and theoretical results in Sections 2 to 4, and then present our empirical results in Section 5. All proofs are deferred to the appendix.

## 2 Scheduling with untrusted signal

We begin by studying a simplified scenario in which the progress bars have a granularity $g = 1$. We assume that there exists some known $\alpha \in [0, 1]$ and unknown $\beta_j \in [0, 1]$ for all $j \in [n]$ such that $\varphi_j(x) = \alpha \cdot \mathbb{1}(\beta_j \le x < 1) + \mathbb{1}(x = 1)$. This model is equivalent to assuming that each job emits a signal after being processed for an unknown fraction $\beta_j$ of its total processing time.

### 2.1 Truthful signals

First, we assume that each job emits a signal exactly after being processed for a fraction $\alpha$ of its total processing time, i.e., $\beta_j = \alpha$. The following theorem describes an optimal algorithm in this setting.

**Theorem 2.1.** *Consider the algorithm that runs Round-Robin over all jobs, and whenever a job emits its signal, it is granted* preferential execution, *i.e. it is run alone until completion. This algorithm achieves a competitive ratio of $1 + \alpha$, which is optimal even for randomized algorithms.*

### 2.2 Untrusted signals

Then, we suppose the signal can be inaccurate: each job $j$ emits its signal after being processed for a fraction $\beta_j \in [0, 1]$ of its total processing time, instead of the announced $\alpha$. We show how to adapt the previous algorithm to maintain robustness under such deviations. A natural baseline is to *naively trust the predictions*, running the previous algorithm for truthful signals described in Theorem 2.1.

**Lemma 2.2.** *Blindly following the signals yields a total completion time of at most*

$$(1 + \alpha)\text{OPT} + \sum_{i=1}^{n}(n - i)(\beta_i - \alpha)p_i + \sum_{i<j}(p_j - p_i) \cdot \mathbb{1}(\beta_j p_j < \beta_i p_i).$$

The theorem above shows that the algorithm is $(1 + \alpha)$-*consistent*, which is the best possible consistency in this setting, and decomposes the additional cost into two components:

- **Signal timing error:** The first sum captures discrepancies on when signals are received. Interestingly, this term can be negative if signals arrive earlier than expected, potentially improving performance in the absence of scheduling inversions.

- **Inversion error:** The second sum can then be interpreted as the total *inversion error*, which is a standard error term in learning-augmented scheduling [LM25a] and captures the increase of the objective due to scheduling jobs in a non-optimal order.

The inversion error in the previous lemma can be upper bounded using the $\ell_1$-norm of the signal timing errors $\sum_{i=1}^{n} |\beta_i - \alpha|$, resulting in the following bound.

**Corollary 2.3.** *Blindly following the signals yields a total completion time of at most* $(1 + \alpha)\mathrm{OPT} + \left(1 + \frac{1}{\alpha}\right) n \sum_{i=1}^{n} |\beta_i - \alpha| p_i$ .

The advantage of this algorithm is that it is straightforward, as it does not even depend on $\alpha$. However, it lacks robustness: its performance can degrade arbitrarily given adversarial predictions. A standard approach of prior work on learning-augmented scheduling to address this issue is to run the algorithm "concurrently" with Round-Robin [PSK18]. Specifically, the consistent algorithm is run at rate $\lambda$, while Round-Robin is run at rate $1 - \lambda$, yielding a consistency of $1+\alpha/\lambda$ and robustness of $2/\lambda$. Nonetheless, our goal is to achieve the *perfect consistency* $1 + \alpha$ while minimizing the robustness.

**Robust algorithm.** We propose the following refined strategy: when a job emits its signal at time $t$, it receives preferential execution for $(1/\alpha - 1)e_i(t)$ units of time, which is exactly to the remaining processing time if the prediction were accurate. If the job finishes during this phase, Round-Robin is resumed. Otherwise, the job is *excluded* from Round-Robin until all other jobs have reached the same processing level or have been completed, alternated with preferential execution when some job emits its signal. In other words, our algorithm runs Shortest Elapsed Time First (SETF) for exploration phases. SETF always processes equally the jobs that have received the least amount of processing so far. This refined strategy corresponds to Algorithm 1 below, parametrized with $\rho = 1$.

While this strategy improves robustness, it suffers from a critical weakness: *brittleness* [EADL24, BP25]. That is, its performance degrades abruptly even for arbitrarily small errors in the signal emission times. We formalize this claim in Proposition A.3, and further illustrate it with experimental results in Figure 2a.

To mitigate this, we introduce Algorithm 1, which satisfies all desired criteria, i.e., perfect consistency, robustness, and smoothness. This algorithm can be reinterpreted as an interpolation between the smooth algorithm from Lemma 2.2 (recovered for $\rho \to 0$) and the robust strategy described above (recovered for $\rho = 1$). Our main result on Algorithm 1 is Theorem 2.4.

---

**Algorithm 1:** $(1 + \alpha)$-consistent, $(1 + 1/\alpha\rho)$-robust algorithm

---

**1** $\mathcal{J} \leftarrow [n]$
**2** **while** $\mathcal{J} \neq \emptyset$ **do**
**3**     Run SETF on $\mathcal{J}$.
**4**     **if** *a job $j$ emits its signal at time $t$* **then**
**5**        Run job $j$ alone for $(1/\alpha\rho - 1) \cdot e_j(t)$ units of time.
**6**        **if** *job $j$ completes* **then**
**7**           $\mathcal{J} \leftarrow \mathcal{J} \setminus \{j\}$

---

**Theorem 2.4.** *Let $\rho \in (0, 1]$. Algorithm 1 is $(1 + \alpha)$-consistent, $(1 + 1/\rho\alpha)$-robust, and if $\rho \in (0, 1)$,*

$$\mathrm{ALG} \leq (1 + \alpha)\mathrm{OPT} + \frac{2n}{\rho(1 - \rho)\alpha^2} \sum_{i=1}^{n} |\beta_i - \alpha| p_i .$$

In Appendix E, we show that these ideas can also be applied in the more general scheduling environment of $m$ *parallel identical machines*. This setting requires tackling additional hurdles, and we give a $(1 + \alpha)$-consistent and $(1 + 1/\alpha)$-robust algorithm as a preliminary proof-of-concept.

Interestingly, while most learning-augmented algorithms exhibit a *consistency-robustness* tradeoff, our algorithm reveals a *smoothness-robustness* tradeoff, governed by the parameter $\rho$. In particular, for $\rho = 1$, the algorithm achieves robustness $1 + 1/\alpha$. We also prove a lower bound on the robustness of any $(1 + \alpha)$-consistent algorithm for this problem. The construction is inspired from [WZ20].

**Theorem 2.5.** *If $\alpha \in (0, 1)$, then any $(1 + \alpha)$-consistent deterministic algorithm has a robustness of at least $1 + \Omega(\sqrt{1/\alpha})$.*

### 2.3 Application to non-clairvoyant scheduling with predictions

Our results for untrusted progress bars can be leveraged to design a learning-augmented algorithm for non-clairvoyant scheduling with predictions on the processing times. Specifically, if the algorithm is provided with a prediction $\pi_j$ of $p_j$ for each job $j \in [n]$, we can frame it in the setting of progress bars with granularity 1 as follows: we fix a parameter $\alpha \in [0, 1]$ and simulate that a job's progress bar jumps to level $\alpha$ when it is processed for $\alpha \pi_j$ units of time. The scheduling is then performed using Algorithm 1 with parameter $\alpha$ for some $\rho \in (0, 1)$. This yields a consistency of $1 + \alpha$ and a robustness of $1 + 1/\rho\alpha$. Moreover, defining $\beta_j = \alpha \pi_j / p_j$, the error term appearing in Theorem 2.4 becomes proportional to $\sum_j |\pi_j - p_j|$.

Notably, this consistency-robustness tradeoff improves upon the best-known algorithms in the learning-augmented scheduling literature. Indeed, the state-of-the-art tradeoff achieves consistency $1/\lambda$ and robustness $\frac{2}{1-\lambda}$ for $\lambda \in (0, 1)$ [PSK18], whereas, with our approach, choosing $\alpha = \frac{1-\lambda}{\rho(1+\lambda)}$ leads to the same robustness $\frac{2}{1-\lambda}$ and consistency $1+\frac{1-\lambda}{\rho(1+\lambda)}$, which can be made arbitrarily close to $\frac{2}{1+\lambda}$ as $\rho \to 1$.

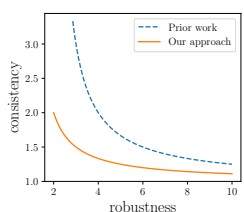

Furthermore, existing algorithms typically rely only on the permutation induced by the predictions on the processing times, whereas our algorithm makes explicit use of the predicted values $\pi_j$. This provides the first separation between algorithms using value-based and permutation-based predictions, two prediction types that were often treated indistinguishably in prior work [PSK18, LM25a].

## 3 Scheduling with untrusted progress bar

We now consider the general setting where each job is equipped with a progress bar of granularity $g \in \mathbb{N}$, which can display progress levels $0 < \alpha^{(1)} < \cdots < \alpha^{(g)} < 1$, with jumps between these levels occurring for actual progress levels $0 \le \beta_j^{(1)} \le \ldots \le \beta_j^{(g)} \le 1$ for all $j \in [n]$.

A naive strategy in this setting is to select a single index $h \in [g]$ and apply Algorithm 1 using only the $h^{\text{th}}$ jump in the progress bar for each job. This yields the guarantees from Theorem 2.4 with parameters $\alpha^{(h)}$ and $(\beta_j^{(h)})_j$. The natural question we ask is whether it is possible to achieve a performance close to that of choosing the best index in hindsight, among the $g$ indices available.

### 3.1 Combining scheduling algorithms

To address this question, we revisit a foundational problem in learning-augmented algorithms: how to combine multiple algorithms to achieve a performance close to the best among them. We approach this problem for scheduling in a general framework, which extends beyond progress bars.

Suppose we are given $g$ algorithms $\mathrm{A}^{(1)}, \ldots, \mathrm{A}^{(g)}$, which may leverage different types of advice or predictions. For each $h \in [g]$ and for all $i \neq j \in [n]$, let $d^{(h)}(i, j)$ denote the delay that job $i$ causes to job $j$ under algorithm $\mathrm{A}^{(h)}$. We assume that each algorithm $\mathrm{A}^{(h)}$ has *computable delays*, meaning that $d^{(h)}(i, j)$ can be determined when both jobs $i$ and $j$ have been completed (i.e., the computation may depend on $p_i, p_j$, the progress bar of both jobs, and possibly other information given during the computation). This assumption holds for most standard scheduling algorithms, whether or not they use advice, since analyzing the total completion time typically relies on studying mutual delays; e.g. Round-Robin satisfies $d^{(h)}(i, j) = \min\{p_i, p_j\}$. Note that it also holds for the algorithm that blindly follows the prediction in the single signal model since $d(i, j)$ can be expressed as a function of $p_i, \beta_i, p_j, \beta_j$ (see the proof of Lemma 2.2).

---

**Algorithm 2:** Combining multiple algorithms $A^{(1)}, \ldots, A^{(g)}$ with computable delays

---

**1** Sample $m$ pairs $(u_k, v_k)_{k=1}^m \sim \binom{n}{2}$.

**2** Run jobs $i \in \{u_k, v_k\}_{k=1}^m$ until completion in any order.

**3** For all $h \in [g]$: compute $a(h) = \sum_{k=1}^m (d^{(h)}(u_k, v_k) + d^{(h)}(v_k, u_k))$.

**4** Let $\hat{h} = \arg\min_h a(h)$. Schedule the remaining jobs using algorithm $A^{(\hat{h})}$.

---

Our combining strategy generalizes the method of [EKMM24], which considers the restricted setting of multiple permutation predictions. Their algorithm samples $m$ pairs of jobs at random, runs them to completion, uses the empirical inversion error to select the best-performing permutation, and then schedules the remaining jobs following that permutation. We extend this idea to general scheduling algorithms with computable delays. After completing the sampled jobs, we compute the cumulative delays induced by each algorithm on this sample. We then schedule the remaining jobs using the algorithm with the lowest empirical total delay. Crucially, our approach works for any algorithms with computable delays, and is not restricted to algorithms with predictions. We give a formal description in Algorithm 2.

**Theorem 3.1.** *Let $A^{(1)}, \ldots, A^{(g)}$ be any deterministic scheduling algorithms having computable delays. Then Algorithm 2 with $m = \frac{1}{8} n^{2/3} (\log g)^{1/3}$ satisfies*

$$\mathbb{E}[\mathrm{ALG}] \leq \min_{h \in [g]} A^{(h)} + \frac{9}{4} n^{5/3} (\log g)^{1/3} \max_{i \in [n]} p_i .$$

This bound immediately generalizes to random instances, or to algorithms using random parameters, as we explain in Appendix B.1. If $\max_i p_i = o(n^{-5/3} \mathrm{OPT})$, then the regret term in Theorem 3.1 becomes $o(\mathrm{OPT})$. This condition is satisfied as soon as a constant fraction of jobs have size at least $\Omega(n^{-\epsilon} \max_i p_i)$ for some $\epsilon < 1/3$, as in that case $\mathrm{OPT} = \Omega(n^{2-\epsilon} \max_i p_i) = \omega(n^{5/3} \max_i p_i)$.

### 3.2 Application to untrusted progress bars

We apply Theorem 3.1 to the setting of untrusted progress bars. For each $h \in [g]$, define $A^{(h)}$ as Algorithm 1 instantiated with $\alpha = \alpha^{(h)}$, ignoring all jumps except the $h^{\text{th}}$ one. Additionally, we combine Round-Robin with the family of algorithms $(A^{(h)})_h$ to guarantee robustness. Thus, we can set $\rho = 0$ for all algorithms $(A^{(h)})_h$, which yields the best smoothness bound (Lemma 2.2). Using Lemma 2.2, Theorem 3.1, and the fact that Round-Robin is 2-competitive, gives the following result.

**Corollary 3.2.** *In the non-clairvoyant scheduling problem with progress bars, suppose each job $i$ emits signals after fractions $\beta_i^{(1)} \leq \ldots \leq \beta_i^{(g)}$ of its total processing time, instead of the announced $\alpha^{(1)} \leq \ldots \leq \alpha^{(g)}$. Then there exists an algorithm with expected total completion time at most*

$$\min \left( 2 \, \mathrm{OPT}, \min_{h \in [g]} A^{(h)} \right) + O \left( n^{5/3} (\log g) \cdot \max_{i \in [n]} p_i \right) ,$$

*where $A^{(h)} \leq (1 + \alpha^{(h)}) \, \mathrm{OPT} + \sum_{i=1}^n (n-i)(\beta_i^{(h)} - \alpha^{(h)}) p_i + \sum_{i<j} (p_j - p_i) \cdot \mathbb{1}(\beta_j^{(h)} p_j < \beta_i^{(h)} p_i)$.*

### 3.3 Application to non-clairvoyant scheduling with predictions

In the context of learning-augmented scheduling, Theorem 3.1 can be used to combine Round-Robin with any consistent algorithm that has computable delays. As an illustration, consider scheduling with a permutation prediction $\sigma$ that predicting the optimal job order. The algorithm that blindly trusts this prediction has a total cost of at most $\mathrm{OPT} + \sum_{i<j} (p_j - p_i) \cdot \mathbb{1}(\sigma(j) < \sigma(i))$ [LM25a], and has computable delays: for all $i \neq j$, the delay that job $i$ incurs to job $j$ is $d(i, j) = p_i \cdot \mathbb{1}(\sigma(i) < \sigma(j))$. Consequently, both algorithms can be combined using Algorithm 2, yielding the following result.

**Corollary 3.3.** *Given a permutation prediction $\sigma$, there exists an algorithm with expected total completion time at most*

$$\min \left( 2 \, \mathrm{OPT}, \mathrm{OPT} + \sum_{i<j} (p_j - p_i) \cdot \mathbb{1}(\sigma(j) < \sigma(i)) \right) + O(n^{5/3} \max_{i \in [n]} p_i) .$$

This result narrows the gap between known upper and lower bounds bounds on the consistency-robustness tradeoff for learning-augmented scheduling. Specifically, in the lower bound of [WZ20], setting $\lambda = 1/\sqrt{n}$ implies that any $(1 + O(1/\sqrt{n}))$-consistent algorithm must have robustness at least $2 + \Omega(1/\sqrt{n})$. In striking contrast, the best upper bound on this tradeoff from prior work is $\left(\frac{1}{\lambda}, \frac{2}{1-\lambda}\right)$ for $\lambda \in (0,1)$ [PSK18], which we improved to $\left(\frac{2}{1+\lambda}, \frac{2}{1-\lambda}\right)$ in Section 2.3. Our approach, for the first time, achieves consistency $1 + o(1)$ and robustness $2 + o(1)$ on instances satisfying $\max_i p_i = o(n^{-5/3}\mathrm{OPT})$, providing a first piece of evidence that near-perfect consistency and robustness could be simultaneously achievable in learning-augmented scheduling.

## 4  Scheduling with stochastic progress bar

**Model.**  The models studied in the previous sections may appear overly pessimistic, as real-world progress bars are not adversarial but instead provide noisy approximations of the true progress [WBB$^+$17, HRB18, XGWC23]. To better capture this, we study a new model of stochastic progress bars that naturally fits applications in resource management and machine learning. For example, consider training a neural network until it reaches a certain accuracy. The training can be seen as a sequence of optimization steps (e.g., gradient descent updates) for which the number of iterations is not known precisely in advance. Yet, at any time, the fraction of the training that has been completed can be estimated from the current validation error, and from the appropriate scaling laws.

Specifically, we consider progress bars $\varphi$ of granularity $g \in \mathbb{N}$, where we model the jumps of the progress bar by the first $g$ jumps of a Poisson process with rate $g$, meaning that the elapsed times between jumps are independent exponential random variables with parameter $g$. Poisson processes are a standard tool for modeling random event arrivals in queueing theory and related areas [Ben57, Wol82, GG96]. Formally, for $j \in [n]$, denote by $\tilde{\beta}_j^{(1)} \leq \cdots \leq \tilde{\beta}_j^{(g)}$ the first $g$ points of a Poisson point process with rate $g$ on $\mathbb{R}^+$. Set $\beta_j^{(h)} = \min(1, \tilde{\beta}_j^{(h)})$ for all $h \in [g]$ and $\beta_j^{(g+1)} = 1$. We define a stochastic progress bar as follows,

$$\varphi_j(x) = \frac{1}{g+1} \sum_{h=1}^{g+1} \mathbb{1}(x \geq \beta_j^{(h)}) \ .$$

**Remark 4.1** (Uniform progress bars). *An alternative model is the one in which $\{\beta_j^{(1)}, \cdots, \beta_j^{(g)}\}$ are drawn i.i.d. and uniformly from $[0,1]$, and then arranged in increasing order. This stochastic setting is sometimes called Binomial point process [Kal17], because, for any $x \in [0,1]$, the value $g\varphi(x)$ is a Binomial random variable, specifically $g\varphi(x) \sim \mathcal{B}(g, x)$. This contrasts with the Poisson point process model defined above, for which $g\varphi(x)$ is a Poisson random variable, specifically $g\varphi(x) \sim \mathcal{P}(gx)$. Both models are closely related via the well-known Poisson approximation, which is a classical technique in probabilistic analysis [MU17], and which essentially states that $\mathcal{B}(g, x)$ and $\mathcal{P}(gx)$ are statistically indistinguishable for small values of $x \in [0,1]$. The Poisson formulation often allows for cleaner proofs, which is why we focus on this model in the theorem statements.*

**Connection to multi-armed bandits.**  Our setting is analogous to a multi-armed bandit problem with $n$ arms [LS20], where arms can be pulled in continuous time instead of discrete rounds, and rewards are generated over time—specifically, each jump of a progress bar from one level to the next is treated as a reward. Short jobs correspond to arms that emit rewards more frequently, aligning with our scheduling objective: complete shorter jobs as quickly as possible.

A key feature of our problem, however, is that the number of jobs decreases over time, resembling the *mortal bandits* setting [CKRU08]. In mortal bandits, algorithms typically rely on an aggressive exploration phase of the currently alive arms, followed by strong exploitation, contrasting with the more gradual exploration-exploitation tradeoff seen in classical algorithms like $\varepsilon$-greedy [SB98] or Upper Confidence Bound (UCB) methods [ACF02]. This intuition carries over to our setting in which the measure of performance is a competitive ratio: when two jobs have similar processing times, it is preferable to commit early to one of the two jobs rather than to waste time estimating which is the shortest of the two.

**Repeated Explore-Then-Commit algorithm.**  Following this intuition, we propose an algorithm that alternates between exploration and exploitation. It runs Round-Robin on all alive jobs until the

displayed progress of one of them reaches a threshold $k/g+1$ (for a carefully chosen parameter $k$), at which point the algorithm fully commits to completing that job. It then resumes Round-Robin on the remaining jobs and iterates this process. Thus, the algorithm alternates between phases of exploration (Round-Robin) and aggressive exploitation (committing to a job); see Algorithm 3. It can also be seen as a variant of Algorithm 1 that ignores all except the $k$-th signals and where $\alpha\rho \to 0$.

---

**Algorithm 3:** Repeated Explore-Then-Commit with threshold $k$

1   $\mathcal{J} \leftarrow [n]$
2   **while** $\mathcal{J} \neq \emptyset$ **do**
3     Round-robin step on $\mathcal{J}$.
4     **if** $\exists j \in \mathcal{J} : X_j(e_j(t)) \geq k/g+1$ **then**
5       Run job $j$ until completion and set $\mathcal{J} \leftarrow \mathcal{J} \setminus \{j\}$.

---

In the next theorem, we prove an upper bound on the competitive ratio of Algorithm 3.

**Theorem 4.2.** *Let $g \geq 12$. For $k = \lceil (g/2)^{2/3} \rceil + 1$, Algorithm 3 has a (expected) competitive ratio of at most $1 + (12/g)^{1/3}$ for minimizing the total completion time with a stochastic progress bar.*

A natural question is whether the convergence rate of $1 + O(g^{-1/3})$ is optimal. We answer this question affirmatively, and establish the asymptotic optimality of our algorithm.

**Theorem 4.3.** *The (expected) competitive ratio of any scheduling algorithm with a Poisson stochastic progress bar of granularity $g \in \mathbb{N}$ is at least $1 + \frac{1}{36}g^{-1/3}$.*

The proof of this lower bound is similar to lower bound arguments for stochastic bandits, yet applied in the different context of scheduling and of competitive analysis. We show that for two jobs with similar processing times, say $p_1 = 1$ and $p_2 = 1 + \tau$ with $\tau = g^{-1/3}$, the processes $e \to X_1(e)$ and $e \to X_2(e)$ are statistically indistinguishable with non-negligible probability if $e < \tau$. Morally, this means that the best thing an algorithm can do is running both jobs in parallel until a progress of $\tau$.

Finally, we also prove an upper bound on the competitive ratio with high probability in Appendix C.3.

## 5   Experiments

We now present empirical experiments that validate our theoretical results. In the experiments, we consider instances with $n = 500$ jobs, where processing times are sampled independently from a Pareto distribution with parameter 1.1. Each point in the figures represents an average over at least 20 independent trials, with standard deviation indicated. In this section, we give a short overview of our findings. An extensive description of our setup and results is given in Appendix D.

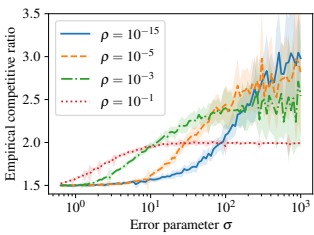 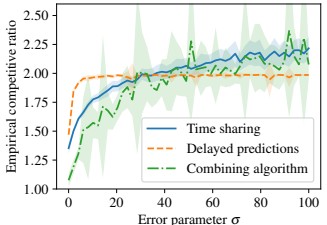 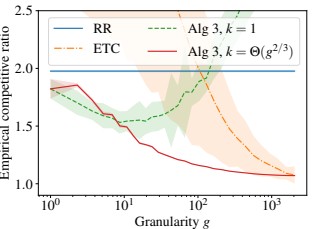

(a) Alg. 1 for different values of $\rho$     (b) Robustification algorithms     (c) Stochastic Progress Bar

Figure 2

In Figures 2a and 2b, the predictions $(\pi_i)_i$ are noisy estimates of the true job sizes $p_i$, with independent Gaussian noise: $\pi_i \sim \mathcal{N}(p_i, \sigma)$. Figure 2a shows the behavior of Algorithm 1, with $\alpha = 0.5$, for various values of $\rho$, where signal emission times are computed as $\beta_i = \max(0, \min(1, \pi_i/p_i))$. As Theorem 2.4 suggests, setting $\rho$ close to 1 improves robustness, but the algorithm's performance is not smooth. On the other hand, smaller $\rho$ values improve smoothness at the expense of robustness.

Figure 2b compares three robustification strategies: *time sharing* [PSK18], *delayed predictions* (Section 2.3), and the *combining algorithm* (Section 3.3). The first two use tuned hyperparameters to ensure the same robustness level. From the figure, delayed predictions appear to guarantee better robustness than time sharing, but the performance is highly sensitive to errors. In contrast, the combining algorithm requires no parameter tuning and achieves both strong robustness and consistency, while keeping good smoothness guarantees. The gap in performance between the combining algorithm and the other strategies becomes more significant for larger values of $n$.

Finally, in Figure 2c, we compare the performance of Algorithm 3 (with $k = 1$ and $k = \Theta(g^{2/3})$), Round-Robin (RR), and a generic Explore-Then-Commit (ETC) algorithm, which runs RR until all jobs reach displayed progress $\Theta(g^{-1/3})$, and then schedules the jobs sequentially in decreasing order of their displayed progress. Our results show that the algorithm of Theorem 4.2 clearly outperforms the other algorithms over the full range of granularities. Moreover, the empirical competitive ratio approaches 1 as $g$ grows, validating the bound of Theorem 4.2.

## 6 Conclusion and open directions

We initiated the study of progress bars in the context of non-clairvoyant scheduling. For both adversarial and stochastic progress bars, we developed algorithms and analyzed their performance guarantees. We believe that this framework is suited to other online problems and that it provides a realistic perspective going beyond worst-case analysis.

Finally, we highlight two open directions. First, in the adversarial single-signal progress bar, there is a gap of $\Theta(\alpha^{-1/2})$ between the upper bound and the lower bound in our analysis of the consistency-robustness tradeoff (cf. Theorem 2.4 and Theorem 2.5). Finding out the exact asymptotic tradeoff is an interesting open problem. Second, in the stochastic progress bar model, our results do not improve over the competitive ratio of 2 when the granularity of the progress bar is very small (specifically, for $g \leq 12$). Particularly natural is the special setting where each job has a single uniformly distributed signal. We believe that even this minimal information is sufficient to break the competitive ratio of 2, with experimental evidence suggesting that the competitive ratio could be below $5/3$.

## Acknowledgments and Disclosure of Funding

We thank the anonymous reviewers for their helpful comments. RC carried out part of this work while at Inria, Paris. AL was supported by the "Humans on Mars Initiative", funded by the Federal State of Bremen and the University of Bremen. JS was supported by the research project "Optimization for and with Machine Learning (OPTIMAL)", funded by the Dutch Research Council (NWO), grant number OCENW.GROOT.2019.015.

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

# A  Proofs of Section 2

## A.1  Proof of Theorem 2.1

In this section, we prove Theorem 2.1, which we restate below for convenience.

**Theorem 2.1.** *Consider the algorithm that runs Round-Robin over all jobs, and whenever a job emits its signal, it is granted* preferential execution*, i.e. it is run alone until completion. This algorithm achieves a competitive ratio of $1 + \alpha$, which is optimal even for randomized algorithms.*

The upper bound of the theorem is a corollary of Lemma 2.2, which we show in the next section, noting that accurate signals imply $\alpha = \beta_j$ for all $j \in [n]$. Hence, here we focus on proving the lower bound. The proof is adapted from [MPT94] and appears concurrently in [GKL+25].

**Lemma A.1.** *For every $\alpha \in [0, 1]$, every randomized algorithm that receives a single accurate signal at $\alpha$ has a competitive ratio of at least $1 + \alpha$ for minimizing the total completion time.*

*Proof.* Using Yao's principle, we fix a deterministic algorithm and consider a randomized instance with $n$ jobs $1, \ldots, n$ with processing times $p_j$ drawn independently from the exponential distribution with scale 1. That is, every job has a size of at least $x$ with probability $e^{-x}$, and $\mathbb{E}[p_j] = 1$.

Fix two arbitrary jobs $i$ and $j$. We compute their expected pairwise delay $\mathbb{E}[d(i, j) + d(j, i)]$ in the algorithm's schedule. To this end, consider for any fixed processing amount $y$ the event $\mathcal{E}(y)$ that neither $i$ nor $j$ has emitted its signal after the algorithm processed a total amount of $y$ on both $i$ and $j$ together. Let $y_i$ denote the part of $y$ used for processing job $i$. Note that

$$\mathbb{P}[\mathcal{E}(y)] = \mathbb{P}[\alpha p_i > y_i] \cdot \mathbb{P}[\alpha p_j > y - y_i] = e^{-y/\alpha} .$$

Therefore, the expected pairwise delay of $i$ and $j$ while both have not yet emitted their signal is equal to $\int_0^\infty \mathbb{P}(\mathcal{E}(y)) \, dy = \alpha$. Furthermore, the expected pairwise delay of $i$ and $j$ while at least one already emitted its signal is equal to

$$\mathbb{E}[\min\{(1 - \alpha)p_i, (1 - \alpha)p_j\}] = (1 - \alpha) \int_0^\infty \mathbb{P}(p_i > x) \cdot \mathbb{P}(p_j > x) \, dx = \frac{1}{2}(1 - \alpha) .$$

Thus, $\mathbb{E}[d(i, j) + d(j, i)] = \frac{1}{2}(1 + \alpha)$. Using (1), we conclude that $\mathbb{E}[\text{ALG}] \geq n + \sum_{j=1}^n \sum_{i=1}^{j-1} \frac{1}{2}(1 + \alpha)$.

We now study the optimal objective value OPT for this instance. In this scenario, an optimal schedule schedules job in non-decreasing order of their processing times. Thus, the expected pairwise delay of $i$ and $j$ in the optimal schedule is equal to $\mathbb{E}[\min\{p_i, p_j\}] = \frac{1}{2}$. Hence, $\mathbb{E}[\text{OPT}] = n + \sum_{j=1}^n \sum_{i=1}^{j-1} \frac{1}{2}$.

We conclude that the competitive ratio is at least

$$\frac{n + \sum_{j=1}^n \sum_{i=1}^{j-1} \frac{1}{2}(1 + \alpha)}{n + \sum_{j=1}^n \sum_{i=1}^{j-1} \frac{1}{2}} ,$$

which approaches $1 + \alpha$ as $n$ goes to $\infty$, and thus, implies the statement. $\qquad\square$

## A.2  Proof of Lemma 2.2

Recall that "blindly following the signals" refers to the algorithm of Theorem 2.1. That is, to the algorithm that runs round-robin over all jobs and, whenever a job $j$ emits a signal, runs only this job until completion before returning to round-robin.

**Lemma 2.2.** *Blindly following the signals yields a total completion time of at most*

$$(1 + \alpha)\text{OPT} + \sum_{i=1}^n (n - i)(\beta_i - \alpha)p_i + \sum_{i<j} (p_j - p_i) \cdot \mathbb{1}(\beta_j p_j < \beta_i p_i).$$

*Proof.* Let $i < j \in [n]$, i.e., $p_i \leq p_j$. Recall that the algorithm processes both jobs with the same rate until one of them emits its signal. Hence, the term $\min\{\beta_i p_i, \beta_j p_j\}$ describes the amount of processing each of the two jobs receive until one of the emits its signal.

If $\beta_i p_i < \beta_j p_j$ then job $i$ emits its signal first, when both jobs have been executed each for $\beta_i p_i$ units of time, then it is executed until completion. Thus the mutual delay caused by the two jobs to each other is

$$d(i,j) + d(j,i) = p_i + \beta_i p_i = (1+\alpha)p_i + (\beta_i - \alpha)p_i .$$

On the other hand, if $\beta_i p_i > \beta_j p_j$, then we obtain similarly that

$$\begin{aligned}
d(i,j) + d(j,i) &= \beta_j p_j + p_j \\
&= (1+\alpha)p_i + (\beta_j p_j - \alpha p_i) + (p_j - p_i) \\
&\leq (1+\alpha)p_i + (\beta_i - \alpha)p_i + (p_j - p_i)
\end{aligned}$$

where we used in the last inequality that $\beta_i p_i > \beta_j p_j$. In both cases, we can write that

$$d(i,j) + d(j,i) \leq (1+\alpha)p_i + (\beta_i - \alpha)p_i + (p_j - p_i)\mathbb{1}(\beta_i p_i > \beta_j p_j) ,$$

and it follows that

$$\begin{aligned}
\text{ALG} &= \sum_{i=1}^{n} p_i + \sum_{i<j} d(i,j) + d(j,i) \\
&\leq \sum_{i=1}^{n} p_i + \sum_{i<j} ((1+\alpha)p_i + (\beta_i - \alpha)p_i + (p_j - p_i)\mathbb{1}(\beta_i p_i > \beta_j p_j)) \\
&= \sum_{i=1}^{n} p_i + (1+\alpha)\sum_{i=1}^{n}(n-i)p_i + \sum_{i=1}^{n}(n-i)(\beta_i - \alpha)p_i + \sum_{i<j}(p_j - p_i)\mathbb{1}(\beta_j p_j < \beta_i p_i) \\
&\leq (1+\alpha)\text{OPT} + \sum_{i=1}^{n}(n-i)(\beta_i - \alpha)p_i + \sum_{i<j}(p_j - p_i)\mathbb{1}(\beta_j p_j < \beta_i p_i) .
\end{aligned}$$

$\square$

The inversion error in the previous lemma can be upper bounded using the absolute signal timing errors, resulting in the following bound.

**Corollary A.2.** *Blindly following the signals yields a total completion time of at most*

$$(1+\alpha)\text{OPT} + \left(1 + \frac{1}{\alpha}\right) n \sum_{i=1}^{n} |\beta_i - \alpha|p_i .$$

*Proof.* For all $i < j$, if $\beta_j p_j < \beta_i p_i$ then

$$\begin{aligned}
p_j - p_i &\leq p_j - p_i + \frac{\beta_i p_i - \beta_j p_j}{\alpha} \\
&= p_j - \frac{\beta_j p_j}{\alpha} + \frac{\beta_i p_i}{\alpha} - p_i \\
&\leq \frac{1}{\alpha}|\beta_j - \alpha|p_j + \frac{1}{\alpha}|\beta_i - \alpha|p_i .
\end{aligned}$$

It follows that

$$\begin{aligned}
\sum_{i<j}(p_j - p_i)\mathbb{1}(\beta_j p_j < \beta_i p_i) &\leq \sum_{i<j} \frac{1}{\alpha}|\beta_j - \alpha|p_j + \frac{1}{\alpha}|\beta_i - \alpha|p_i \\
&= \sum_{i=1}^{n}\sum_{j\neq i} \frac{1}{\alpha}|\beta_i - \alpha|p_i \\
&= \frac{n-1}{\alpha}\sum_{i=1}^{n}|\beta_i - \alpha|p_i .
\end{aligned}$$

We deduce from Lemma 2.2 that

$$\text{ALG} \leq (1+\alpha)\text{OPT} + (n-1)\sum_{i=1}^{n} |\beta_i - \alpha|p_i + \frac{n-1}{\alpha}\sum_{i=1}^{n} |\beta_i - \alpha|p_i$$

$$\leq (1+\alpha)\text{OPT} + \left(1 + \frac{1}{\alpha}\right)n\sum_{i=1}^{n} |\beta_i - \alpha|p_i .$$

$\square$

### A.3 Proof of the brittleness of Algorithm 1 with $\rho = 1$

**Proposition A.3.** *If $\alpha < 1$, then for all $\varepsilon > 0$, there is an instance of job sizes $(p_j)_{j\in[n]}$ and signal emission fractions $(\beta_j)_{j\in[n]}$ satisfying $\sum_{j=1}^{n} |\beta_j - \alpha| \leq \varepsilon$, for which Algorithm 1 with $\rho = 1$ satisfies*

$$\frac{\text{ALG}}{\text{OPT}} \geq 2 .$$

*This means that an arbitrarily small error in the signal emission times suffices to deviate from the consistency bound $1 + \alpha$.*

*Proof.* Let $\varepsilon > 0$, $m \geq 1$, and $\delta \leq \frac{\varepsilon}{2m}$. Consider an instance with $n = 2m$ jobs, with sizes given by $p_j = 1$ for $j \leq m$ and $p_j = 1/\alpha$ for $j > m$. For all $j \in [2m]$, let $\beta_j = \alpha - \delta$, so that the total deviation from the announced value $\alpha$ is $\sum_j |\beta_j - \alpha| = 2m\delta \leq \varepsilon$.

Running Algorithm 1 with parameter $\rho = 1$, the initial phase is Round-Robin execution until each job has been processed for $\alpha - \delta$ units of time. At this point, the jobs with $j \leq m$ emit their signals by definition of $\beta_j$. To resolve simultaneous emissions, assume infinitesimal perturbations to ensure that signal times are distinct. These $m$ jobs, each with size 1, then receive preferential execution one after the other, until reaching an elapsed time of $\beta_j/\alpha = 1 - \delta/\alpha$. In particular, none of these jobs completed yet, as their size is 1.

At this stage, the second set of $m$ jobs ($j > m$), each of size $1/\alpha$, have received less processing and are now scheduled using Round-Robin, until they reach elapsed times of $1 - \frac{\delta}{\alpha}$ each, at which point they emit their signals and receive preferential execution, until they have an elapsed time of $\beta_j p_j/\alpha = 1/\alpha - \delta/\alpha^2$. This happens at time $m(1 - \delta/\alpha) + m(1/\alpha - \delta/\alpha^2) = m(1 + 1/\alpha)(1 - \delta/\alpha)$. Since no job has been completed by this time, the sum of completion times incurred by the algorithm satisfies:

$$\text{ALG} \geq 2m \cdot m(1 + 1/\alpha)(1 - \delta/\alpha) = 2m^2(1 + 1/\alpha)(1 - \delta/\alpha) .$$

On the other hand,

$$\text{OPT} = \sum_{j=1}^{m} j + \sum_{j=1}^{m}(m + j/\alpha) = m^2\left(\frac{3}{2} + \frac{1}{2\alpha}\right) + o(m^2) .$$

Therefore

$$\frac{\text{ALG}}{\text{OPT}} \geq (1 - \delta/\alpha) \cdot \frac{4\alpha + 4}{3\alpha + 1} - o(1) .$$

Note that, for $\alpha \in [0, 1)$, we have $\frac{4\alpha+4}{3\alpha+1} > 2$. Hence, for $\delta$ sufficiently small and $m$ sufficiently large, the lower bound above becomes larger than 2, which concludes the proof. $\square$

### A.4 Proof of Theorem 2.4

**Theorem 2.4.** *Let $\rho \in (0, 1]$. Algorithm 1 is $(1+\alpha)$-consistent, $(1 + 1/\rho\alpha)$-robust, and if $\rho \in (0, 1)$,*

$$\text{ALG} \leq (1+\alpha)\text{OPT} + \frac{2n}{\rho(1-\rho)\alpha^2}\sum_{i=1}^{n} |\beta_i - \alpha|p_i .$$

*Proof.* As in other proofs, we assume that $p_1 \leq \ldots \leq p_n$. Let $k = \frac{1}{\rho\alpha} > \frac{1}{\alpha}$, and $i < j \in [n]$. We will analyze the mutual delay between jobs $i$ and $j$, depending on their job sizes and the times when their signals are emitted. To this end, we distinguish between multiple cases and, for each case prove a robustness bound of the form $d(i,j) + d(j,i) \leq (1+k)p_i$ and an error-dependent bound of

$$d(i,j) + d(j,i) \leq (1+\alpha)p_i + \frac{k+1}{\alpha - 1/k} \left( |\beta_i - \alpha|p_i + |\beta_j - \alpha|p_j \right).$$

**Case 1** $k\beta_i \leq 1$**:** Note that $k\beta_i \leq 1$ implies that $i$ either does not finish during its preferential execution ($k\beta_i < 1$) or the preferential execution suffices to *exactly* finish $i$ ($k\beta_i = 1$). For this case, we can observe that proving $d(i,j) + d(j,i) \leq (1+k)p_i$ immediately implies an error-dependent bound, just by using $\beta_i \leq \frac{1}{k}$:

$$d(i,j) + d(j,i) \leq (1+k)p_i = (1+\alpha)p_i + (k-\alpha)p_i \leq (1+\alpha)p_i + \frac{k-\alpha}{\alpha - 1/k}(\alpha - \beta_i)p_i$$

$$\leq (1+\alpha)p_i + \frac{k+1}{\alpha - 1/k} \left( |\beta_i - \alpha|p_i + |\beta_j - \alpha|p_j \right).$$

Hence, it suffices to show $d(j,i) + d(i,j) \leq (1+k)p_i$. We do so by distinguishing two sub-cases.

**Case 1.1** $\beta_j p_j \geq C_i$**:** If $j$ emits its signal after $i$ completes, then the SETF-part of the algorithm ensures that the delay is at most $d(j,i) + d(i,j) \leq 2p_i \leq (1+k) \cdot p_i$.

**Case 1.2** $\beta_j p_j < C_i$**:** For $j$ to emit its signal before $i$ completes, we must have $\beta_j p_j \leq p_i$. The maximum delay of $i$ caused by $j$ is $d(j,i) = \max\{k\beta_j p_j, p_i\} \leq \max\{kp_i, p_i\} = kp_i$, where the inequality follows from $\beta_j p_j \leq p_i$. Note that the first term $k\beta_j p_j$ in the maximum is the maximum delay in case that $j$ completes during its preferential execution, which is composed of a delay of $\beta_j p_j$ before $j$ emits and a delay of $(k-1)\beta_j p_j$ during the preferential treatment. Since $d(i,j) \leq p_i$ holds trivially, we can conclude with $d(j,i) + d(i,j) \leq (1+k)p_i$.

**Case 2** $k\beta_i > 1$ **and** $\beta_j p_j > \beta_i p_i$**:** In this case, $i$ emits its signal before $j$ due to $\beta_j p_j > \beta_i p_i$. Furthermore, $i$ completes during its preferential execution due to $k\beta_i > 1$. Thus, the delay of $i$ by $j$ is only caused during the parallel execution before $i$ emits its signal. Hence $d(j,i) \leq \beta_i p_i$. Since $d(i,j) \leq p_i$ holds trivially, we conclude $d(j,i) + d(i,j) \leq (1+\beta_i)p_i \leq (1+k)p_i$. This immediately implies an error-dependent bound:

$$d(i,j) + d(j,i) \leq (1+\beta_i)p_i \leq (1+\alpha)p_i + |\beta_i - \alpha|p_i$$

$$\leq (1+\alpha)p_i + \frac{k+1}{\alpha - 1/k} \left( |\beta_i - \alpha|p_i + |\beta_j - \alpha|p_j \right).$$

**Case 3** $k\beta_i > 1$ **and** $\beta_j p_j \leq \beta_i p_i$**:** In this case, $j$ emits before or at the same time as $i$ ($\beta_j p_j \leq \beta_i p_i$) and $i$ completes during its preferential execution ($k\beta_i > 1$). We start with the general inequality

$$p_j - p_i \leq p_j - \frac{\beta_j p_j}{\alpha} + \frac{\beta_i p_i}{\alpha} - p_i \leq \frac{1}{\alpha}|\beta_j - \alpha|p_j + \frac{1}{\alpha}|\beta_i - \alpha|p_i, \qquad (2)$$

which will be useful below. Next, we distinguish between three sub-cases.

**Case 3.1** $k\beta_j \geq 1$**:** Job $j$ is the first to emit its signal and is then executed until completion due to $k\beta_j \geq 1$. Thus, $d(i,j) = \beta_j p_j$ as $i$ only delays $j$ during the parallel execution before $j$ emits its signal. Furthermore, $d(j,i) = p_j$ as $j$ finishes before $i$ during the preferential execution. Using $\beta_j p_j \leq \beta_i p_i \leq p_i$, we obtain

$$d(i,j) + d(j,i) = \beta_j p_j + p_j \leq (1+k)\beta_j p_j \leq (1+k)p_i.$$

Furthermore, $\beta_j p_j \le \beta_i p_i$ and (2) give

$$
\begin{aligned}
d(i,j) + d(j,i) &= \beta_j p_j + p_j \\
&= (1+\alpha)p_i + (\beta_j p_j - \alpha p_i) + (p_j - p_i) \\
&\le (1+\alpha)p_i + (\beta_i - \alpha)p_i + \frac{1}{\alpha}|\beta_i - \alpha|p_i + \frac{1}{\alpha}|\beta_j - \alpha|p_j \\
&\le (1+\alpha)p_i + \left(1 + \frac{1}{\alpha}\right)|\beta_i - \alpha|p_i + \frac{1}{\alpha}|\beta_j - \alpha|p_j \\
&\le (1+\alpha)p_i + \frac{k}{\alpha - 1/k}|\beta_i - \alpha|p_i + \frac{1}{\alpha}|\beta_j - \alpha|p_j \\
&\le (1+\alpha)p_i + \frac{k+1}{\alpha - 1/k}\left(|\beta_i - \alpha|p_i + |\beta_j - \alpha|p_j\right).
\end{aligned}
$$

where we used in the second to last inequality that

$$
\alpha + 1 \le \alpha + \frac{1}{\alpha} \le k + \frac{1}{k} \le k + \frac{1}{k} + \frac{1}{\alpha k}\,,
$$

which is equivalent to $\left(\alpha - \frac{1}{k}\right)\left(1 + \frac{1}{\alpha}\right) \le k$.

**Case 3.2** $k\beta_j < 1$ **and** $k\beta_j p_j > \beta_i p_i$**:** Job $j$ again emits its signal first and receives preferential execution, but fails to terminate during the preferential execution as $k\beta_j < 1$. Since $k\beta_j p_j > \beta_i p_i$, job $i$ emits its signal before catching up with the elapsed time of $j$, and is then executed until completion. Thus $d(i,j) = p_i$ and $d(j,i) = k\beta_j p_j$. We have

$$
d(i,j) + d(j,i) = p_i + k\beta_j p_j \le p_i + k\beta_i p_i \le (1+k)p_i\,.
$$

Additionally, it follows from $k\beta_j < 1$ that

$$
\begin{aligned}
d(i,j) + d(j,i) &\le p_i + k\beta_j p_j \\
&= (1+\alpha)p_i + k\beta_j p_j - \alpha p_i \\
&= (1+\alpha)p_i + (k\beta_j p_j - p_j) + (p_i - \alpha p_i) + (p_j - p_i) \\
&\le (1+\alpha)p_i + (1-\alpha)p_i + (p_j - p_i) \\
&\le (1+\alpha)p_i + \frac{1-\alpha}{\alpha - 1/k}(\alpha - \beta_j)p_j + (p_j - p_i)\,,
\end{aligned}
$$

where we used in the last inequality that $k\beta_j < 1$ and $p_i \le p_j$. Finally, (2) gives

$$
\begin{aligned}
d(i,j) + d(j,i) &\le (1+\alpha)p_i + \frac{1-\alpha}{\alpha - 1/k}(\alpha - \beta_j)p_j + \frac{1}{\alpha}|\beta_i - \alpha|p_i + \frac{1}{\alpha}|\beta_j - \alpha|p_j \\
&= (1+\alpha)p_i + \frac{1}{\alpha}|\beta_i - \alpha|p_i + \frac{2-\alpha}{\alpha - 1/k}|\beta_j - \alpha|p_j \\
&\le (1+\alpha)p_i + \frac{k+1}{\alpha - 1/k}\left(|\beta_i - \alpha|p_i + |\beta_j - \alpha|p_j\right).
\end{aligned}
$$

**Case 3.3** $k\beta_j < 1$ **and** $k\beta_j p_j \le \beta_i p_i$**:** In this case, the elapsed time of job $i$ reaches that of job $j$ before job $i$ emits its signal, both jobs are then processed equally until job $i$ emits its signal, then it receives preferential execution until completion. Hence $d(i,j) = p_i$ and $d(j,i) = \beta_i p_i$. It follows immediately that

$$
d(i,j) + d(j,i) = (1+\beta_i)p_i \le (1+k)p_i\,,
$$

and

$$
d(i,j) + d(j,i) = (1+\alpha)p_i + (\beta_i - \alpha)p_i \le (1+\alpha)p_i + \frac{k+1}{\alpha - 1/k}\left(|\beta_i - \alpha|p_i + |\beta_j - \alpha|p_j\right).
$$

From analyzing all the possible cases, we deduce that, for all values of $p_i \le p_j$ and $\beta_i, \beta_j$, the mutual delay $d(i,j) + d(j,i)$ is at most $(1+k)p_i$. Using (1), it follows that

$$
\mathrm{ALG} = \sum_{i=1}^n p_i + \sum_{i<j}(d(i,j)+d(j,i)) \le \sum_{i=1}^n p_i + (1+k)\sum_{i<j} p_i \le (1+k)\mathrm{OPT}\,,
$$

which proves that the robustness ratio is at most $1 + 1/\rho\alpha$ if $k = 1/\rho\alpha$ for some $\rho \in (0, 1]$.

We will now deduce the smoothness bound. To lighten the notation, we denote by $\Delta_i = |\beta_i - \alpha|p_i$ for all $i \in [n]$. In the case-analysis above, we have already shown the following error-dependent bound for all $i < j$:

$$d(i, j) + d(j, i) \leq (1 + \alpha)p_i + \frac{k + 1}{\alpha - 1/k}(\Delta_i + \Delta_j) \ ,$$

which yields after summation that

$$
\begin{aligned}
\text{ALG} &= \sum_{i=1}^{n} p_i + \sum_{i<j}(d(i, j) + d(j, i)) \\
&\leq \sum_{i=1}^{n} p_i + (1 + \alpha)\sum_{i<j} p_i + \frac{k + 1}{\alpha - 1/k}\sum_{j=1}^{n}\sum_{i=1}^{j-1}(\Delta_i + \Delta_j) \\
&\leq (1 + \alpha)\text{OPT} + \frac{k + 1}{\alpha - 1/k}\left(\sum_{i=1}^{n}(n - i)\Delta_i + \sum_{j=1}^{n}(j - 1)\Delta_j\right) \\
&\leq (1 + \alpha)\text{OPT} + \frac{k + 1}{\alpha - 1/k}n\sum_{i=1}^{n}\Delta_i \ .
\end{aligned}
$$

Finally, it holds for $k = 1/(\rho\alpha)$ that

$$\frac{k + 1}{\alpha - 1/k} = \frac{1/(\rho\alpha) + 1}{(1 - \rho)\alpha} = \frac{1 + \rho\alpha}{\rho(1 - \rho)\alpha^2} \leq \frac{2}{\rho(1 - \rho)\alpha^2} \ ,$$

which concludes the proof. $\qquad\square$

Even for $\rho = 1$, we have no tight example for the robustness bound of Algorithm 1. The best example that we have shows that the robustness of this algorithm is in $1 + \Omega(\sqrt{1/\alpha})$. Thus, we leave as open question whether the robustness analysis of the algorithm can be improved.

## A.5  Proof of Theorem 2.5

**Theorem 2.5.** *If $\alpha \in (0, 1)$, then any $(1 + \alpha)$-consistent deterministic algorithm has a robustness of at least $1 + \Omega(\sqrt{1/\alpha})$.*

*Proof.* Fix a deterministic $\alpha$-clairvoyant algorithm. Consider an instance $I_1$ composed of $n$ jobs with unit processing times. Let $t$ denote the first time when a job completes. We assume that $1 = e_1 \geq \ldots \geq e_n$, where $e_j = e_j(t)$ denotes the total progress of job $j$ until time $t$. Thus, job 1 completes at time $t$. We assume the algorithm does not idle, and have $t = \sum_{i=1}^{n} e_i$. Note that the best strategy for any algorithm at time $t$ is to finish the jobs in the order of their index as $1 - e_1 \leq \ldots \leq 1 - e_n$. Since the algorithm is by assumption $(1 + \alpha)$-competitive, it must hold that

$$nt + \sum_{j=1}^{n}(n - j + 1)(1 - e_j) \leq \text{ALG} \leq (1 + \alpha)\text{OPT} = (1 + \alpha)\frac{n(n + 1)}{2} \ .$$

Since $t = \sum_{j=1}^{n} e_j$, the above can only be true if

$$\sum_{j=1}^{n}(j - 1)e_j \leq \alpha\frac{n(n + 1)}{2} \ . \tag{3}$$

Let $\varepsilon > 0$ be sufficiently small. We consider another instance $I_2$ with processing times $p_1 = 1$ and $p_j = e_j + \varepsilon$ for every job $2 \leq j \leq n$. The jobs emits their signal until time $t$ as they have done in instance $I_1$. Note that these signals may not emit after an $\alpha$-fraction of the jobs are done, which is

possible because we are in the robustness case. Thus, the algorithm processes $I_1$ and $I_2$ equivalently until time $t$, as it is deterministic. We have for $\varepsilon \to 0$ that

$$\frac{\text{ALG}}{\text{OPT}} \geq \frac{n\left(\sum_{j=1}^{n} e_j\right)}{\sum_{j=1}^{n} j \cdot e_j} = \frac{n\left(e_1 + \sum_{j=2}^{n} e_j\right)}{e_1 + \sum_{j=2}^{n} j \cdot e_j} \ .$$

Now, by considering this ratio as a function of $e_2, \ldots, e_n$ subject to $\sum_{j=2}^{n} e_j = t - e_1$ and $e_2 \geq \ldots \geq e_n$, note that it is minimized if and only if $e_2 = \ldots = e_n = \frac{t-e_1}{n-1}$ for all jobs $j$. Thus, the above is at least

$$\frac{nt}{e_1 + \frac{1}{2}(n+2)(t-e_1)} \ .$$

Since this ratio is minimized if $t$ is as large as possible, and the left-hand side of (3) is equal to $n\frac{t-e_1}{2}$, at this minimum (3) must be tight and we have that $t = \alpha(n+1) + e_1$. Thus, the above is at least

$$\frac{n(\alpha(n+1) + e_1)}{e_1 + \frac{1}{2}(n+2) \cdot \alpha(n+1)} = \frac{n + \alpha n(n+1)}{1 + \frac{1}{2}\alpha(n+1)(n+2)} \ .$$

Finally, choosing $n = \Theta(\sqrt{1/\alpha})$ implies that the robustness ratio of the algorithm is at least

$$\Omega\left(\frac{n + \alpha n^2}{\alpha n^2}\right) = \Omega(n) = \Omega(\sqrt{1/\alpha}) \ ,$$

which concludes the proof. $\qquad\square$

While our algorithm does not match this lower bound, it is possible to improve the robustness by relaxing the consistency guarantee. This can be achieved by selecting a parameter $\alpha' \in [\alpha, 1]$ and running Algorithm 1 with $\alpha'$ in place of $\alpha$, using $\rho = \alpha'/\alpha$. Specifically, if a job $j$ emits a signal at time $t$, the algorithm waits until the job has been processed for $\frac{\alpha'}{\alpha} e_j(t)$ before treating the signal as emitted. This modification yields a degraded consistency of $1 + \alpha'$, and improved robustness of $1 + 1/\alpha'$.

## B  Proofs of Section 3

**Theorem 3.1.** *Let* $A^{(1)}, \ldots, A^{(g)}$ *be any deterministic scheduling algorithms having computable delays. Then Algorithm 2 with* $m = \frac{1}{8}n^{2/3}(\log g)^{1/3}$ *satisfies*

$$\mathbb{E}[\text{ALG}] \leq \min_{h \in [g]} A^{(h)} + \frac{9}{4}n^{5/3}(\log g)^{1/3} \max_{i \in [n]} p_i \ .$$

*Proof.* Let $p_{\max} = \max_{i \in [n]} p_i$. Since $p_i \in [0, p_{\max}]$ for all $i \in [n]$, the total delay caused by the first phase of the algorithm, where the jobs $\{u_k, v_k\}_{k=1}^{m}$ are completed, is at most $2mn \cdot p_{\max}$. Algorithm $A^{(\hat{h})}$ is then used to schedule the remaining jobs, with a total objective value of at most $A^{(\hat{h})}$, since having some jobs completed in the first phase can only improve the objective function. Therefore,

$$\text{ALG} \leq A^{(\hat{h})} + 2mn \cdot p_{\max} \ . \tag{4}$$

For all $h \in [g]$ and $i \neq j \in [n]$, we denote by $M^{(h)}(i,j) = d^{(h)}(i,j) + d^{(h)}(j,i)$ the mutual delay caused by jobs $i, j$ to each other when scheduled with algorithm $A^{(h)}$. For all $h \in [g]$, recall that $a(h) = \sum_{k=1}^{m}(d^{(h)}(u_k, v_k) + d^{(h)}(v_k, u_k))$ as defined in Algorithm 2, and note that

$$\mathbb{E}[a(h)] = m\mathbb{E}[M^{(h)}(u_1, v_1)] = \frac{2m}{n(n-1)} \sum_{j=1}^{n} \sum_{i=1}^{j-1} M^{(h)}(i,j) \ .$$

Let $Z(h) := a(h) - \mathbb{E}[a(h)]$ and $h^* = \arg\min A^{(h)}$. It holds that

$$
\begin{aligned}
A^{(\hat{h})} &= \sum_{i=1}^{n} p_i + \sum_{j=1}^{n} \sum_{i=1}^{j-1} M^{(\hat{h})}(i,j) \\
&= \sum_{i=1}^{n} p_i + \frac{n(n-1)}{2m} \mathbb{E}[a(\hat{h})] \\
&= \sum_{i=1}^{n} p_i + \frac{n(n-1)}{2m} a(\hat{h}) - \frac{n(n-1)}{2m} Z(\hat{h}) \\
&\leq \sum_{i=1}^{n} p_i + \frac{n(n-1)}{2m} a(h^*) - \frac{n(n-1)}{2m} Z(\hat{h}) \\
&= \sum_{i=1}^{n} p_i + \frac{n(n-1)}{2m} \mathbb{E}[a(h^*)] + \frac{n(n-1)}{2m} Z(h^*) - \frac{n(n-1)}{2m} Z(\hat{h}) \\
&\leq A^{(h^*)} + \frac{n(n-1)}{2m} \left( \max_{h \in [g]} \{Z(h)\} + \max_{h \in [g]} \{-Z(h)\} \right) .
\end{aligned}
$$

Combining this with Equation (4), and by definition of $h^*$, we obtain in expectation that

$$
\mathbb{E}[\text{ALG}] \leq \min_{h \in [g]} A^{(h)} + \frac{n(n-1)}{2m} \left( \mathbb{E}[\max_{h \in [g]} \{Z(h)\}] + \mathbb{E}[\max_{h \in [g]} \{-Z(h)\}] \right) + 2mn \cdot p_{\max} .
$$

Let us now bound $\mathbb{E}[\max_{h \in [g]} \{Z(h)\}]$ and $\mathbb{E}[\max_{h \in [g]} \{-Z(h)\}]$. We have that $a(h)$ is the sum of $m$ independent random variables $(M^{(h)}(u_k, v_k))_{k=1}^{m}$, all bounded in $[0, 2p_{\max}]$: $M^{(h)}(i,j) = d^{(h)}(i,j) + d^{(h)}(j,i) \leq p_i + p_j \leq 2p_{\max}$. Thus, $a(h)$ is a subgaussian random variable with variance factor $\sigma^2 = \sum_{k=1}^{m} (2p_{\max} - 0)^2 / 4 = m p_{\max}^2$ [KT23]. Therefore, $\max_h Z(h)$ is the maximum of $g$ subgaussian random variables all with variance factor $m p_{\max}^2$ (although not independent), hence, using the result of [KT23], we have

$$
\mathbb{E}[\max_{h \in [g]} Z(h)] \leq p_{\max} \sqrt{2m \log g} .
$$

Similarly, we obtain the same upper bound on $\mathbb{E}[\max_{h \in [g]} \{-Z(h)\}]$. We deduce that

$$
\mathbb{E}[\text{ALG}] \leq \min_{h \in [g]} A^{(h)} + \frac{n^2 p_{\max}}{2m} \sqrt{2m \log g} + 2mn \cdot p_{\max} = \min_{h \in [g]} A^{(h)} + \left( n^2 \sqrt{\frac{\log g}{2m}} + 2mn \right) p_{\max} .
$$

Finally, for $m = \frac{1}{8} n^{2/3} (\log g)^{1/3}$, the above bound becomes

$$
\mathbb{E}[\text{ALG}] \leq \min_{h \in [g]} A^{(h)} + \frac{9}{4} n^{5/3} (\log g)^{1/3} p_{\max} ,
$$

which proves the theorem. $\qquad \square$

**Remark B.1.** *If we are unable to compute $d^{(h)}(i,j)$ exactly for all $h$ after completing two jobs $i$ and $j$, but can instead compute an upper bound $\bar{d}^{(h)}(i,j) \geq d^{(h)}(i,j)$, then the upper bound stated in the theorem still applies by replacing $A^{(h)}$ with $\bar{A}^{(h)}$ for each $h$, where*

$$
\bar{A}^{(h)} = \sum_{i=1}^{n} p_i + \sum_{i<j} \left( \bar{d}^{(h)}(i,j) + \bar{d}^{(h)}(j,i) \right) .
$$

*This guarantees that the combining algorithm achieves a total completion time at most the minimum of these theoretical upper bounds, up to the same regret term.*

### B.1 Random instances and randomized algorithms

The result of [EKMM24] holds with high probability. While we can also prove an upper bound w.h.p., our upper bound on the expectation is particularly useful as it allows an extension to random job sizes and randomized algorithms, as we show below.

**Generalization to random instances.** If the job sizes are random, then the same choice of $m = \frac{1}{8}n^{2/3}(\log g)^{1/3}$ in Algorithm 2 gives

$$\mathbb{E}[\mathrm{ALG}] \leq \min_{h \in [g]} \mathbb{E}[\mathrm{A}^{(h)}] + \left( \frac{9}{4} n^{5/3} (\log g)^{1/3} \right) \mathbb{E}[\max_{i \in [n]} p_i] \, .$$

Indeed, assuming that the job sizes are random, we obtain by Theorem 3.1, conditionally on the job sizes, that

$$\mathbb{E}[\mathrm{ALG} \mid p_1, \ldots, p_n] \leq \min_{h \in [g]} \mathrm{A}^{(h)} + \left( \frac{9}{4} n^{5/3} (\log g)^{1/3} \right) \max_{i \in [n]} p_i \, ,$$

hence, using that $\mathbb{E}[\min_{h \in [g]} \mathrm{A}^{(h)}] \leq \min_{h \in [g]} \mathbb{E}[\mathrm{A}^{(h)}]$, we find

$$\mathbb{E}[\mathrm{ALG}] \leq \min_{h \in [g]} \mathbb{E}[\mathrm{A}^{(h)}] + \left( \frac{9}{4} n^{5/3} (\log g)^{1/3} \right) \mathbb{E}[\max_{i \in [n]} p_i] \, .$$

For example, if the job sizes are independently sampled from an exponential distribution with parameter 1, then

$$\mathbb{E}[\mathrm{ALG}] \leq \min_{h \in [g]} \mathbb{E}[\mathrm{A}^{(h)}] + \frac{9}{4} (n^{5/3} \log n)(\log g)^{1/3}$$

$$= \min_{h \in [g]} \mathbb{E}[\mathrm{A}^{(h)}] + o(\mathbb{E}[\mathrm{OPT}]) \, .$$

**Generalization to randomized algorithms.** The result can be easily generalized to algorithms using random parameters that can be sampled before starting the execution of the jobs (for e.g. choosing a random permutation). Assuming that each algorithm $\mathrm{A}^{(h)}$ uses a random vector $\xi^{(h)}$ as a parameter, that can be sampled before starting the algorithm, then the bound of Theorem 3.1 gives

$$\mathbb{E}[\mathrm{ALG} \mid \xi^{(1)}, \ldots, \xi^{(g)}] \leq \min_{h \in [g]} \mathrm{A}^{(h)}(\xi^{(h)}) + \frac{9}{4} n^{5/3} (\log g)^{1/3} \max_{i \in [n]} p_i \, ,$$

hence

$$\mathbb{E}[\mathrm{ALG}] \leq \mathbb{E}[\min_{h \in [g]} \mathrm{A}^{(h)}(\xi^{(h)})] + \frac{9}{4} n^{5/3} (\log g)^{1/3} \max_{i \in [n]} p_i$$

$$\leq \min_{h \in [g]} \mathbb{E}[\mathrm{A}^{(h)}(\xi^{(h)})] + \frac{9}{4} n^{5/3} (\log g)^{1/3} \max_{i \in [n]} p_i \, .$$

## C  Proofs of Section 4

### C.1  Proof of Theorem 4.2

**Theorem 4.2.** *Let $g \geq 12$. For $k = \lceil (g/2)^{2/3} \rceil + 1$, Algorithm 3 has a (expected) competitive ratio of at most $1 + (12/g)^{1/3}$ for minimizing the total completion time with a stochastic progress bar.*

*Proof.* For all $j \in [n]$, let $\tau_j$ denote the total processing time allocated to job $j$ by the point at which its progress bar reaches the threshold $\frac{k}{g+1}$.

Let $i \neq j$. If $\tau_i < \tau_j$, then both jobs $i, j$ are executed with similar rates until job $i$ emits its signal, after which it is executed until completion. Therefore, $d(i, j) = p_i$ and $d(j, i) = \tau_i$.

Assume that $p_1 \leq \ldots \leq p_n$, and let $i < j \in [n]$. It holds that

$$d(i, j) + d(j, i) = (p_i + \tau_i)\mathbb{1}(\tau_i \leq \tau_j) + (p_j + \tau_j)\mathbb{1}(\tau_j < \tau_i)$$

$$= p_i \mathbb{1}(\tau_i \leq \tau_j) + p_j \mathbb{1}(\tau_j < \tau_i) + \left( \tau_i \mathbb{1}(\tau_i \leq \tau_j) + \tau_j \mathbb{1}(\tau_j < \tau_i) \right)$$

$$= p_i + \min(\tau_i, \tau_j) + (p_j - p_i)\mathbb{1}(\tau_j < \tau_i) \, ,$$

Recalling that $\mathbb{E}[\tau_i] = k/\mu_i = \frac{k}{g}p_i$, and by independence of $\tau_i$ and $\tau_j$, we obtain directly that $\mathbb{E}[\min(\tau_i, \tau_j)] \leq \frac{k}{g}\min(p_i, p_j) = \frac{k}{g}p_i$, and it follows that

$$\frac{1}{p_i}\mathbb{E}[d(i,j) + d(j,i)] \leq 1 + \frac{k}{g} + (p_j/p_i - 1)\mathbb{P}(\tau_j < \tau_i)$$

$$\leq 1 + \frac{k}{g} + (p_j/p_i - 1) \cdot \frac{k - 1/2}{\sqrt{\pi(k-1)}} \int_{\frac{p_j - p_i}{p_j + p_i}}^{1} \left(1 - x^2\right)^{k-1} dx \qquad (5)$$

$$\leq 1 + \frac{k}{g} + \frac{k - 1/2}{\sqrt{\pi(k-1)}} \sup_{r \geq 1}\left\{(r - 1)\int_{\frac{r-1}{r+1}}^{1}\left(1 - x^2\right)^{k-1} dx\right\} \qquad (6)$$

$$\leq 1 + \frac{k}{g} + \frac{k - 1/2}{\sqrt{\pi(k-1)}} \cdot \frac{1}{k - 2} \qquad (7)$$

$$\leq 1 + \frac{k}{g} + \frac{1}{\sqrt{k-1}} \; .$$

where (5) follows from Lemma C.1 taking $m = k - 1 \geq 3$, (6) is obtained immediately by taking $r = p_j/p_i$, (7) follows from Lemma C.2, and the last inequality holds because $k \geq 4$.

Finally, by definition of $k$, we obtain that

$$\frac{1}{p_i}\mathbb{E}[d(i,j) + d(j,i)] \leq 1 + \frac{(g/2)^{2/3} + 2}{g} + \frac{1}{(g/2)^{1/3}} = 1 + \frac{2}{g} + \frac{3}{(4g)^{1/3}} \leq 1 + \left(\frac{12}{g}\right)^{1/3} \; ,$$

and it follows from (1) that

$$\mathbb{E}[\mathrm{ALG}] = \sum_{i=1}^{n} p_i + \sum_{j=1}^{n}\sum_{i=1}^{j-1}\mathbb{E}[d(i,j) + d(j,i)]$$

$$\leq \left(\sum_{i=1}^{n} p_i + \sum_{j=1}^{n}\sum_{i=1}^{j-1} p_i\right) + \left(\frac{12}{g}\right)^{1/3}\sum_{j=1}^{n}\sum_{i=1}^{j-1} p_i \leq \mathrm{OPT} + \left(\frac{12}{g}\right)^{1/3}\mathrm{OPT} \; .$$

$\square$

**Lemma C.1.** *If $\tau_1, \tau_2$ are independent random variables, following respectively Gamma distributions with parameters $(m + 1, \mu_1)$ and $(m + 1, \mu_2)$, then*

$$\mathbb{P}(\tau_2 < \tau_1) \leq \frac{m + 1/2}{\sqrt{\pi m}} \int_{\frac{\mu_1 - \mu_2}{\mu_1 + \mu_2}}^{1}\left(1 - x^2\right)^{m} dx \; .$$

*Proof.* For $i \in \{1, 2\}$, the probability density function of $\tau_i$ is

$$t \in [0, \infty) \mapsto \frac{\mu_i^{m+1}}{m!}t^m e^{-\mu_i t} \; ,$$

hence

$$\mathbb{P}(\tau_2 < \tau_1) = \int_{0}^{\infty}\frac{\mu_1^{m+1}}{m!}t^m e^{-\mu_1 t}\int_{0}^{t}\frac{\mu_2^{m+1}}{m!}u^m e^{-\mu_2 u}du\,dt$$

$$= \int_{0}^{\infty}\frac{\mu_1^{m+1}}{m!}t^m e^{-\mu_1 t}\int_{0}^{1}\frac{\mu_2^{m+1}}{m!}(tu)^m e^{-\mu_2 ut}t\,du\,dt$$

$$= \frac{(\mu_1\mu_2)^{m+1}}{(m!)^2}\int_{0}^{1}u^m\int_{0}^{\infty}t^{2m+1}e^{-(\mu_1 + \mu_2 u)t}dt\,du$$

$$= \frac{(\mu_1\mu_2)^{m+1}}{(m!)^2}\int_{0}^{1}u^m\int_{0}^{\infty}\frac{t^{2m+1}}{(\mu_1 + \mu_2 u)^{2m+1}}e^{-t}\frac{dt}{\mu_1 + \mu_2 u}du$$

$$= \frac{(\mu_1\mu_2)^{m+1}}{(m!)^2}\int_{0}^{1}\frac{u^m}{(\mu_1 + \mu_2 u)^{2m+2}}\left(\int_{0}^{\infty}t^{2m+1}e^{-t}dt\right)du \; .$$

The inner integral corresponds to the Gamma function evaluated at $2m + 2$, which equals $(2m + 1)!$ as $2m + 2$ is integer, hence

$$\mathbb{P}(\tau_2 < \tau_1) = \frac{(2m + 1)!}{(m!)^2}(\mu_1\mu_2)^{m+1} \int_0^1 \frac{u^m}{(\mu_1 + \mu_2 u)^{2m+2}} du \ .$$

Making the substitution $x = \frac{2\mu_1}{\mu_1 + \mu_2 u} - 1$, we have that $dx = -\frac{2\mu_1\mu_2}{\mu_1 + \mu_2 u} du$ and $1 - x^2 = \frac{4\mu_1\mu_2 u}{(\mu_1 + \mu_2 u)^2}$, hence

$$\mathbb{P}(\tau_2 < \tau_1) = \frac{(2m + 1)!}{(m!)^2} \cdot \frac{1/2}{4^m} \int_{\frac{\mu_1 - \mu_2}{\mu_1 + \mu_2}} (1 - x^2)^m dx \ .$$

Finally, using a well-known inequality on the central binomial coefficient [Mer23], we obtain that

$$\frac{(2m + 1)!}{(m!)^2} = (2m + 1)\binom{2m}{m} \le (2m + 1)\frac{4^m}{\sqrt{\pi m}} \ ,$$

and it follows that

$$\mathbb{P}(\tau_2 < \tau_1) \le \frac{m + 1/2}{\sqrt{\pi m}} \int_{\frac{\mu_1 - \mu_2}{\mu_1 + \mu_2}}^1 (1 - x^2)^m dx \ .$$

$\square$

**Lemma C.2.** *If $m \ge 3$, then it holds that*

$$\sup_{r \ge 1} \left\{ (r - 1) \int_{\frac{r-1}{r+1}}^1 (1 - x^2)^m dx \right\} \le \frac{1}{m - 1} \ .$$

*Proof.* With a variable change $u = \frac{r-1}{r+1}$ we obtain that

$$\sup_{r \ge 1} \left\{ (r - 1) \int_{\frac{r-1}{r+1}}^1 (1 - x^2)^m dx \right\} = \sup_{u \in [0,1)} \left\{ \frac{2u}{1 - u} \int_u^1 (1 - x^2)^m dx \right\} \ ,$$

Define for all $u \in [0, 1)$ the function $f(u) = \frac{2u}{1-u} \int_u^1 (1 - x^2)^m dx$. We will show the upper bound separately for $u > 1/2$ and $u \le 1/2$

**Upper bound for large $u$.** Let $u > 1/2$. Then, $f(u)$ can be simply upper bounded as follows

$$f(u) \le \frac{2u}{1 - u}(1 - u^2)^m \int_u^1 dx = 2u(1 - u^2)^m \ ,$$

With an immediate derivative analysis we have that $u \mapsto 2u(1 - u^2)^m$ is decreasing on $[\frac{1}{\sqrt{2m+1}}, 1]$. In particular, it holds that $\frac{1}{\sqrt{2m+1}} < \frac{1}{2}$ for $m \ge 3$, hence $u \mapsto 2u(1 - u^2)^m$ is maximal on $[1/2, 1]$ for $u = 1/2$, which gives

$$\forall u \ge \frac{1}{2} : \qquad f(u) \le 2u(1 - u^2)^m \le (3/4)^m \le \frac{1}{m - 1} \ . \tag{8}$$

**Upper bound for small $u$.** Consider now $u \le 1/2$. It holds that

$$\int_u^1 (1 - x^2)^m dx \le \int_u^1 e^{-mx^2} dx \le \int_u^\infty e^{-mx^2} dx$$

$$\le \int_u^\infty \frac{x}{u} e^{-mx^2} dx = \frac{1}{2mu} \int_u^\infty 2mx e^{-mx^2} dx = \frac{e^{-mu^2}}{2mu} \ ,$$

hence

$$f(u) \le \frac{2u}{1 - u} \cdot \frac{e^{-mu^2}}{2mu} \le \frac{1}{m} \cdot \sup_{t \in [0, \frac{1}{2}]} \frac{e^{-mt^2}}{1 - t} \ .$$

$t \mapsto \frac{e^{-mt^2}}{1-t}$ attains its maximum on $[0, 1/2]$ for $t_m = \frac{1}{2} - \frac{1}{2}\sqrt{1 - \frac{2}{m}}$, hence

$$\sup_{t \in [0, \frac{1}{2}]} \frac{e^{-mt^2}}{1-t} = \frac{e^{-mt_m^2}}{1-t_m} \leq \frac{2}{1 + \sqrt{1 - \frac{2}{m}}} \leq \frac{2}{2 - \frac{2}{m}} = \frac{m}{m-1} \ ,$$

and we deduce that

$$\forall u \leq \frac{1}{2} : \qquad f(u) \leq \frac{1}{m} \cdot \frac{m}{m-1} = \frac{1}{m-1} \ . \tag{9}$$

Combining (8) and (9), we conclude that $\sup_{u \in [0,1)} f(u) \leq \frac{1}{m-1}$. $\qquad\square$

## C.2  Proof of Theorem 4.3

We start by stating two classical results from information theory on the Kullback-Leibler (KL) divergence and total variation (TV) distance.

**Proposition C.3** (see e.g. [CT06])**.** *The KL divergence between two Poisson random variables is bounded as follows*

$$D_{KL}(\mathcal{P}(\lambda), \mathcal{P}(\mu)) = \lambda \log\left(\frac{\lambda}{\mu}\right) - (\lambda - \mu) \leq \frac{(\lambda - \mu)^2}{\mu} \ ,$$

*where we used for the inequality $\forall x \geq -1 : \ln(1 + x) \leq x$.*

**Proposition C.4** (Pinsker's inequality, see e.g. [CT06])**.**

$$D_{TV}(\mathcal{P}(\lambda), \mathcal{P}(\mu)) \leq \sqrt{\frac{1}{2} D_{KL}(\mathcal{P}(\lambda), \mathcal{P}(\mu))}.$$

We now apply these results to Poisson random variables with specific parameterization.

**Lemma C.5.** *Let $\tau, g \in \mathbb{R}^+$, we have*

$$D_{KL}\left(\mathcal{P}(g\tau), \mathcal{P}\left(g\tau/(1+\tau)\right)\right) \leq g\tau^3 \ ,$$

*and by Pinsker's inequality, the TV distance between the two distributions is bounded by $\sqrt{\frac{1}{2}g\tau^3}$.*

*Proof.* Using the Proposition C.3 on the divergence between Poisson random variables,

$$D_{KL}\left(\mathcal{P}(g\tau), \mathcal{P}\left(g\tau/(1+\tau)\right)\right) \leq g\tau\left(1 - \frac{1}{1+\tau}\right)^2 \leq g\tau^3,$$

where we use $\forall x \geq 0 : \frac{1}{1+x} \geq 1 - x$. $\qquad\square$

**Lemma C.6.** *Let $\tau \in \mathbb{R}$. We consider two Poisson point processes $X$ and $Y$, with fixed rates. We denote by $X(\tau) \in \mathbb{N}$ the value of the process $X$ at time $\tau$ and by $X(\leq \tau)$ the value of the process on the interval $[0, \tau]$, i.e., $X(\leq \tau) : [0, \tau] \to \mathbb{N}$. Then, it holds that*

$$D_{TV}(X(\leq \tau), Y(\leq \tau)) = D_{TV}(X(\tau), Y(\tau)).$$

*Proof Sketch.* Any coupling of $X(\leq \tau)$ and $Y(\leq \tau)$ induces a coupling of $X(\tau)$ and $Y(\tau)$. Thus, $D_{TV}(X(\tau), Y(\tau)) \leq D_{TV}(X(\leq \tau), Y(\leq \tau))$. For the other inequality, consider a coupling $(X, Y)$ of random variables such that $X \sim X(\tau)$ and $Y \sim Y(\tau)$ and $\mathbb{P}(X \neq Y) = D_{TV}(X(\tau), Y(\tau))$, as well as $U_1, U_2, \ldots$ a list of uniform random variables in $[0, \tau]$. Then define

$$X(t) = \#\{i \in \{1, \ldots, X\} : U_i \leq t\}, \text{ and}$$
$$Y(t) = \#\{i \in \{1, \ldots, Y\} : U_i \leq t\}.$$

Note that if $X = Y$ then, forall $t \leq \tau, X(t) = Y(t)$. Also, it is a standard observation that $X : t \in [0, \tau] \to X(t)$ and $Y : t \in [0, \tau] \to Y(t)$ are two coupled Poisson point processes with the desired intensities. This shows the reverse inequality $D_{TV}(X(\leq \tau), Y(\leq \tau)) \leq D_{TV}(X(\tau), Y(\tau))$. $\qquad\square$

**Lemma C.7.** *Let $\tau \in (0,1)$. Consider an instance with two jobs $\{1,2\}$ with processing times $\{1, 1+\tau\}$. We say that a scheduling algorithm $\mathcal{A}$ has a preference for job 1 if an elapsed time of $\tau$ is attained on job 1 before being attained on job 2. Let q be the probability that $\mathcal{A}$ has a preference for the longest job. Then, the competitive ratio of $\mathcal{A}$ is at least*

$$1 + \frac{q}{3}\tau - \frac{2}{9}\tau^2.$$

*Proof.* For the instance considered, we always have $\text{OPT} = 3 + \tau$. If the algorithm has a preference for the longest job, then $\text{ALG} \geq 3 + 2\tau$ (consider the two options: either the algorithm goes on and finishes the longest job first, or the algorithm finishes the shortest job first but was delayed by the longest job). Denoting by $q$ the probability that the algorithm has a preference for the longest job, we thus have $\mathbb{E}(\text{ALG}) \geq 3 + (1+q)\tau$. Therefore, the competitive ratio is at least

$$\mathbb{E}\left(\frac{\text{ALG}}{\text{OPT}}\right) \geq \frac{3 + (1+q)\tau}{3+\tau} \geq \left(1 + \frac{1+q}{3}\tau\right)\left(1 - \frac{1}{3}\tau\right) \geq 1 + \frac{q}{3}\tau - \frac{2}{9}\tau^2,$$

where we used $\forall x \geq -1 : \frac{1}{1+x} \geq 1 - x$ and $q \leq 1$. $\qquad\square$

**Lemma C.8.** *Consider the instance with two jobs $\{1,2\}$ of processing times $\{1, 1+\tau\}$, assigned at random (i.e., $\mathbb{P}((p_1, p_2) = (1, 1+\tau)) = 1/2$ and $\mathbb{P}((p_1, p_2) = (1+\tau, 1)) = 1/2$). For any scheduling algorithm $\mathcal{A}$ (deterministic or randomized) the probability $q$ that $\mathcal{A}$ has a preference for the longest job is at least equal to $1/2 - d$, where $d = D_{TV}(\mathcal{P}(g\tau), \mathcal{P}(g\tau/(1+\tau)))$.*

*Proof.* We consider $X^S$ the Poisson process of the shortest job (rate 1) and $X^L$ the Poisson process of the longest job (rate $1/(1+\tau)$). We consider two pairs of Poisson point processes on $[0, \tau]$, with rates 1 and $\frac{1}{1+\tau}$, further denoted by $(X^S, X^L)$ and $(\bar{X}^S, \bar{X}^L)$ that are coupled in a way such that under event $A$, with $\mathbb{P}(A) = 1 - 2d$, we have $X^S = \bar{X}^L$ and $X^L = \bar{X}^S$. Note that this coupling can be defined by considering two independent couplings of $X^S$ and $\bar{X}^L$ and of $X^L$ and $\bar{X}^S$, letting $d = D_{TV}(\mathcal{P}(g\tau), \mathcal{P}(g\tau/(1+\tau)))$, and using the union bound.

We then introduce the random variable $B \in \{S, L\}$ saying if job 1 is short (S) or long (L). Note that $B$ is sampled independently at random from all the other random variables. We also introduce the notation $S^c = L$ and $L^c = S$.

The variables $(X_1, X_2)$, corresponding to the progress bars of jobs 1 and 2 until elapsed time of $\tau$, are obtained from $(B, X^S, X^L)$ by letting $(X_1, X_2) = (X^B, X^{B^c})$. Any scheduling algorithm $\mathcal{A}$ induces a (possibly random) function $f$ such that $f(X_1, X_2) \in \{S, L\}$ which outputs $S$ if 1 was preferred by algorithm $\mathcal{A}$ or $L$ if 2 was preferred by algorithm $\mathcal{A}$. The probability that the longest job by algorithm $\mathcal{A}$ is preferred is equal to

$$q = \mathbb{P}_{(B, X^S, X^L)}(f(X^B, X^{B^c}) = B^c). \tag{10}$$

Conversely, the probability that the shortest job is preferred is equal to

$$1 - q = \mathbb{P}_{(B, X^S, X^L)}(f(X^B, X^{B^c}) = B),$$
$$= \mathbb{P}_{(B, X^S, X^L)}(f(X^{B^c}, X^B) = B^c), \tag{11}$$

where we used the fact that $(B, X^S, X^L)$ has the same law as $(B^c, X^S, X^L)$.

We now condition on the event $A$, of probability $1-2d$, for which we can use, $X^B, X^{B^c} = \bar{X}^{B^c}, \bar{X}^B$, and thus $f(X^B, X^{B^c}) = f(\bar{X}^{B^c}, \bar{X}^B)$, which implies

$$\mathbb{E}(\mathbb{1}(A)\mathbb{1}(f(X^B, X^{B^c}) = B^c)) = \mathbb{E}(\mathbb{1}(A)\mathbb{1}(f(\bar{X}^{B^c}, \bar{X}^B) = B^c)). \tag{12}$$

We conclude by noting that the left-hand side is smaller than $q$ by (10) and that the right-hand side is greater than $1 - q - 2d$ by (11) and a union bound. We get that,

$$q \geq 1 - q - 2d$$

and thus,

$$q \geq 1/2 - d.$$

$\qquad\square$

**Theorem 4.3.** *The (expected) competitive ratio of any scheduling algorithm with a Poisson stochastic progress bar of granularity $g \in \mathbb{N}$ is at least $1 + \frac{1}{36} g^{-1/3}$.*

*Proof.* Let $\tau = \frac{1}{4} g^{-1/3}$, which satisfies $\sqrt{\frac{1}{2} g \tau^3} = 2^{-7/2} \leq \frac{1}{4}$. We consider the instance composed of two jobs $\{1, 2\}$ of lengths permuted at random from $\{1, 1 + \tau\}$. By Lemma C.8, the scheduling algorithm has a preference for the longest job with probability $q \geq \frac{1}{2} - d$, where $d = D_{TV}(\mathcal{P}(g\tau), \mathcal{P}(g\tau/(1 + \tau)))$ satisfies $d \leq \sqrt{\frac{1}{2} g \tau^3}$, by Lemma C.5. Thus, by Lemma C.7, the competitive ratio is at least $1 + \frac{q}{3} \tau - \frac{2}{9} \tau^2 \geq 1 + \frac{1}{12} \tau - \frac{1}{18} \tau \geq 1 + \frac{1}{36} \tau$, where we used $q \geq \frac{1}{4}$ and $\tau \leq \frac{1}{4}$. $\qquad \square$

## C.3   Bound with high probability

In this section, we study the tail risk of the competitive ratio of our Algorithm 2. While high-probability bounds for the competitive ratio is relatively, this approach was also recently investigated by [DIL$^+$24] for the ski rental problem. We start with two classical concentration lemmas on Poisson random variables and on Poisson processes.

**Lemma C.9.** *Let $\lambda \in \mathbb{R}$ and $\epsilon \in [0, 1]$. For $X \sim \mathcal{P}(\lambda)$, we have that*

$$\mathbb{P}(X \geq (1 + \epsilon)\lambda) \leq h_1(\epsilon)^\lambda, \quad \text{where} \quad h_1(\epsilon) = \frac{e^\epsilon}{(1 + \epsilon)^{1 + \epsilon}} \leq \exp(-\epsilon^2/3),$$

$$\mathbb{P}(X \leq \lambda/(1 + \epsilon)) \leq h_2(\epsilon)^\lambda, \quad \text{where} \quad h_2(\epsilon) = \left( \frac{1 + \epsilon}{e^\epsilon} \right)^{\frac{1}{1 + \epsilon}} \leq \exp(-\epsilon^2/7).$$

*Proof.* These bounds are also known as multiplicative Chernoff bounds for Poisson variables, and are well-documented, see e.g. [MU17, Th. 5.4]. $\qquad \square$

**Lemma C.10.** *Let $g \in \mathbb{N}$, $\mu \in \mathbb{R}$ and $\epsilon \in [0, 1]$. Consider a Poisson point process $X(\cdot)$ of rate $\mu$, i.e., a stochastic process such that : $t \to X(t)$ is increasing and at any time $t : X(t) \sim \mathcal{P}(t\mu)$. Then, consider $\tau = X^{-1}(g) = \inf\{t \in \mathbb{R} : X(t) \geq g\}$. We have,*

$$\mathbb{P}\left( \mu \notin \left[ \frac{1}{1 + \epsilon} \frac{g}{\tau}, (1 + \epsilon) \frac{g}{\tau} \right] \right) \leq 2 \exp(-\epsilon^2 g/7), \tag{13}$$

*where the probability is taken over the randomness in $\tau$. We shall also call $\hat{\mu} = \frac{g}{\tau}$ the estimator of $\mu$.*

*Proof.* First, we bound,

$$\begin{aligned}
\mathbb{P}\left( \mu < \frac{1}{1 + \epsilon} \frac{g}{\tau} \right) &= \mathbb{P}\left( \tau < \frac{1}{1 + \epsilon} \frac{g}{\mu} \right), \\
&= \mathbb{P}\left( X\left( \frac{1}{1 + \epsilon} \frac{g}{\mu} \right) \geq g \right), \\
&= \mathbb{P}\left( \mathcal{P}\left( \frac{g}{1 + \epsilon} \right) \geq g \right), \\
&\leq h_1(\epsilon)^{g/(1 + \epsilon)} \leq \exp(-\epsilon^2 g/6).
\end{aligned}$$

Then, we bound,

$$\begin{aligned}
\mathbb{P}\left( \mu > (1 + \epsilon) \frac{g}{\tau} \right) &= \mathbb{P}\left( \tau > (1 + \epsilon) \frac{g}{\mu} \right), \\
&= \mathbb{P}\left( X\left( (1 + \epsilon) \frac{g}{\mu} \right) < g \right), \\
&= \mathbb{P}\left( \mathcal{P}((1 + \epsilon)g) < g \right), \\
&\leq h_2(\epsilon)^{(1 + \epsilon)g} \leq \exp(-\epsilon^2 g/7).
\end{aligned}$$

$\qquad \square$

**Theorem C.11.** *The Repeated Explore-then-Commit algorithm with threshold $k \leq g$ has competitive ratio bounded by*

$$1 + 3\epsilon + 4\frac{k}{g}, \tag{14}$$

*with probability at least $1 - 2n \exp(-\epsilon^2 k/7)$, for any $\epsilon \in [0, 1]$.*

*Thus for $\alpha > 0$, in the regime where $\frac{\sqrt{\alpha \log n}}{g^{1/3}} < 1$, taking $k = g^{2/3}$ leads to a competitive ratio in $1 + O\left(\frac{\sqrt{\alpha \log n}}{g^{1/3}}\right)$ with probability at least $1 - O(\frac{1}{n^\alpha})$.*

Note that unlike in Theorem 4.2, the high-probability competitive ratio scales with $n$, in contrast to the expected competitive ratio, which remains independent of $n$. This difference arises because the high-probability guarantee requires concentration bounds that hold for each job $j \in [n]$. We thus expect that a granularity that increases (logarithmically) with $g$ is needed to obtain high probability bounds in such regimes.

*Proof.* We consider jobs ordered by processing times $p_1 \leq \ldots \leq p_n$. For $j \in [n]$, we write $\mu_j = \frac{g}{p_i}$ and denote by $\tau_i \in [0, p_i]$ the instant at which the $k$-th signal is observed for job $j$.

We shall focus on the the event $E$ in which $\forall j \in [n] : \mu_j \in \left[\frac{1}{1+\epsilon} \frac{k}{\tau_j}, (1+\epsilon)\frac{k}{\tau_j}\right]$. We note that by union bound, and by applying Lemma C.10, we have that $\mathbb{P}(E^c) \leq 2n \exp(-\epsilon^2 k/7)$.

We now bound the competitive ratio under event $E$. For all jobs $j$, we note that $\pi_j = \frac{g}{k}\tau_j$ is a $(1+\epsilon)$ estimate of $p_j$. We also decompose the completion time $C_j$ as follows (1) the time spent finishing $p_j$ (2) the time spent doing round robin on other arms, which equals $\sum_{i \neq j} \min\{\tau_i, \tau_j\}$, (3) the time spent finishing other jobs $\sum_{i:\tau_i < \tau_j} (p_i - \tau_i)$. We can also equivalently decompose as follows (2') the time spent on jobs finished before $j$, $\sum_{i:\tau_i < \tau_j} p_i$ and (3') the time spent on other jobs that will finish after $j$, $\sum_{i:\tau_i > \tau_j} \tau_j$. We start by bounding the contribution of (3'),

$$\sum_{j=1}^n \sum_{i=1}^n \mathbb{1}[\tau_i > \tau_j] \cdot \tau_j \leq \sum_{j=1}^n \sum_{i=1}^n \mathbb{1}[i \neq j] \cdot \min\{\tau_i, \tau_j\} \leq 2 \sum_{j=1}^n \sum_{i=1}^{j-1} \min\{\tau_i, \tau_j\}$$

$$\leq 2 \sum_{j=1}^n \sum_{i=1}^{j-1} (1+\epsilon)\frac{k}{g} \min\{p_i, p_j\}$$

$$\leq 2(1+\epsilon)\frac{k}{g} \sum_{j=1}^n \sum_{i=1}^{j-1} p_i$$

$$\leq 2(1+\epsilon)\frac{k}{g}\text{OPT}. \tag{15}$$

We then look at the contribution due to (2') under $E$, and we denote by $\sigma$ the permutation for which the values of $\tau_{\sigma(i)}$ are ordered $\tau_{\sigma(1)} \leq \ldots \leq \tau_{\sigma(n)}$,

$$\sum_{j=1}^n \sum_{i:\tau_i < \tau_j} p_i \leq (1+\epsilon) \sum_{j=1}^n \sum_{i:\tau_i < \tau_j} \pi_i = (1+\epsilon) \sum_{j=1}^n \sum_{i=1}^{j-1} \pi_{\sigma(i)}$$

$$\leq (1+\epsilon)^2 \sum_{j=1}^n \sum_{i=1}^{j-1} p_i$$

where the last inequality comes from $\forall j : \pi_{\sigma(j)} \leq (1+\epsilon)p_j$ under event $E$, since at least $j$ elements in $\{\pi_i : i \in [n]\}$ are smaller than $(1+\epsilon)p_j$ (specifically $\pi_1, \ldots, \pi_j$). Therefore, the combined contribution of (1) and (2') is at most

$$\sum_{i=1}^n p_i + \sum_{j=1}^n \sum_{i:\tau_i < \tau_j} p_i \leq (1+\epsilon)^2\text{OPT}. \tag{16}$$

We conclude by assembling (15) and (16) that, under event $E$, the competitive ratio is bounded by

$$(1 + \epsilon)^2 + 2(1 + \epsilon)\frac{k}{g} \leq 1 + 3\epsilon + 4\frac{k}{g}.$$

The last claim of the theorem statement follows from taking $\epsilon = \sqrt{\alpha \log n} \frac{k}{g}$, assuming that this value is in $[0, 1]$. $\qquad\square$

## D  Experiments

We give a more detailed overview of our experimental setup and results. Unless stated otherwise, we consider instances with $n = 500$ jobs, where processing times are sampled independently from a Pareto distribution with shape parameter $1.1$. This distribution is commonly used in literature to model job sizes in scheduling problems [PSK18, LM25a, BP24].

**Algorithm 1, robustness vs smoothness.**   In the first set of experiments, we compare the performance of Algorithm 1 with a fixed parameter $\alpha = 0.5$ and varying values of $\rho$. As established in Theorem 2.4, the parameter $\rho$ controls the tradeoff between robustness and smoothness. Predictions are generated as described in Section 5.

The two figures below illustrate the ratio of the algorithm's cost to that of OPT as a function of the error parameter $\sigma$, where $\sigma^2$ denotes the variance of the generated predictions. Results are shown for $\rho \in \{10^{-15}, 10^{-5}, 10^{-3}, 10^{-1}\}$. The left figure displays this ratio for $\sigma \in [0, 150]$, while the right figure presents the same metric on a logarithmic scale for $\sigma \in [5 \times 10^{-1}, 10^3]$.

As shown in Theorem 2.4, all values of $\rho$ ensure the same level of consistency, namely $1 + \alpha = 3/2$. However, larger values of $\rho$ improve robustness at the expense of smoothness. This tradeoff is particularly evident when comparing $\rho = 10^{-15}$ with $\rho = 10^{-1}$. For instance, the left figure illustrates that with $\rho = 10^{-1}$, the performance ratio increases abruptly from the consistency bound of $1.5$ to a higher value of $2$ for moderate values of $\sigma$. This effect becomes even more pronounced for larger values of $\rho$; specifically, for $\rho = 1$, the algorithm exhibits brittle behavior, as formally established in Proposition A.3. Larger values of $\rho$ ensure a better smoothness, hence a better overall performance when the prediction error is small, but weaker robustness guarantees, attained when the prediction error is important. The robustness values can be compared more easily in the right figure.

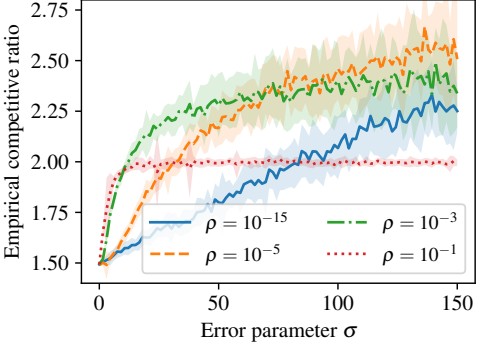
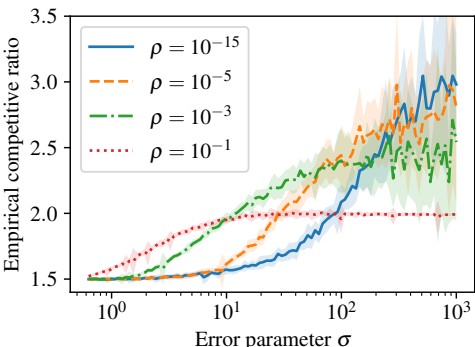

Figure 3: Algorithm 1 with different values of $\rho$, $n = 500$, $\sigma \in [0, 150]$

Figure 4: Algorithm 1 with different values of $\rho$, $n = 500$, $\sigma \in [5 \times 10^{-1}, 10^3]$

**Robustification strategies.**   In the second set of experiments, we evaluate and compare various *robustification strategies*, which are *time sharing* [PSK18], *delayed predictions* (Section 2.3), and the *combining algorithm* (Section 3.3). as shown in Theorem 3.1, the performance of the *combining algorithm* incurs a regret term that diminishes as the sample size $n$ grows. To highlight this behavior, we compare the three robustification strategies for different values of $n$. Predictions are again generated according to Section 5, and the hyperparameters of the time-sharing and delayed-predictions strategies are chosen to guarantee the same level of robustness $3$.

The figures below reveal that for different tested values of $n$, using delayed predictions yields a better robustness on the considered instances of job sizes. However, this strategy lacks smoothness

compared to time-sharing. for $n = 50$, the combining algorithm performs similarly to time-sharing. However, as $n$ increases, the difference between the two approaches becomes more apparent. Notably, for $n = 1000$, the combining algorithm achieves nearly perfect consistency (ratio approaching 1) and robust performance on the tested instances. A key advantage of the combining algorithm is that it automatically balances consistency and robustness, without requiring manual tuning of hyperparameters to adjust this tradeoff.

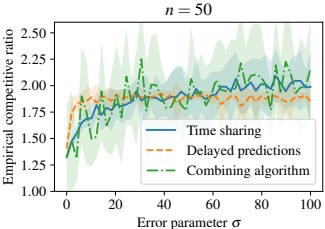 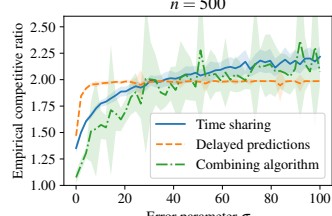 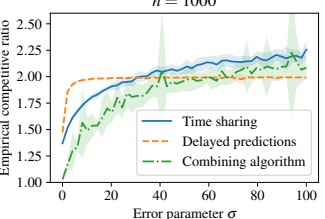

Figure 5: Robustification strategies, for $n = 50$

Figure 6: Robustification strategies, for $n = 500$

Figure 7: Robustification strategies, for $n = 1000$

**Stochastic progress bar.** For the stochastic experiments, which are given in Figure 2c, we average over 50 instances. For each instance, and for each considered value of $g$, we sample for each job independently progress bar signals, and then simulate the algorithms.

# E   Single Untrusted Signal on Multiple Machines

In this section, we provide preliminary results on the non-clairvoyant scheduling problem with a single untrusted signal on *multiple identical machines*. In this setting, we are given $m$ identical machines. At any time $t$, each of the $m$ machines can process at most one job, and each job $j$ can be processed on at most one machine. Using the possibility of preempting jobs arbitrarily often, the machines can simulate the parallel execution of multiple jobs with a lower processing rate. More precisely, we can re-interpret the problem in the following way: At each time $t$, we assign a rate $R_j^t$ with $0 \le R_j^t \le 1$ to each not yet completed job $j$. The assigned rates have to satisfy $\sum_{j \in [n]} R_j^t \le m$ (for the sake of convenience, we assume that already completed jobs have a rate of zero) to not exceed the processing capacities of the machines. Note that the assigned rates do not actually specify on which machine to schedule the jobs. This is without loss of generality as we can use McNaughton's wrap-around-rule [McN59] to transform these rates to an actual schedule that processes any job at any point in time on at most one machine. In this formulation of the problem, the total processing that a job $j$ receives until time $t$ (also called the *elapsed time* of $j$ at $t$) is given by $e_j(t) = \int_0^t R_j^{t'} \, dt'$. The completion time $C_j$ of a job $j$ is still the earliest time $t'$ with $e_j(t') \ge p_j$, and the objective remains to assign rates to minimize $\sum_{j=1}^n C_j$. As the result of this section, we prove the following theorem.

**Theorem E.1.** *There is a $(1+\alpha)$-consistent and $(1+\frac{1}{\alpha})$-robust for non-clairvoyant scheduling with a single untrusted signal on parallel identical machines.*

The theorem shows that the consistency and robustness of Theorem 2.4 for $\rho = 1$ translates to the setting of identical parallel machines. Questions concerning the smoothness and brittleness remain open for future work. To show the theorem, we first give an algorithm and afterwards separately prove consistency and robustness.

**Algorithm**   Let $J = \{1, \dots, n\}$ be a set of $n$ jobs such that $p_1 \le \dots \le p_n$. We consider the following algorithm, which can be seen as variant of Algorithm 1 that replaces SETF with Round-Robin. The algorithm maintains two job sets $E$ and $S$ and a FIFO queue $Q$. At any time, the algorithm ensures that every unfinished job is either in $E$ or in $Q$. The set $S \subseteq Q$ is the subset of the first (at most) $m$ elements of the queue. In the beginning, we set $E \leftarrow J$. At any time $t$, we assign the following rates:

  (i)  Schedule every job $j$ in $E$ at rate $R_j^t = q_t := \min\{1, (m - |S|)/|E|\}$. If a job $j \in E$ emits a signal at time $t$, set $\rho_j \leftarrow e_j(t)$, remove $j$ from $E$ and enqueue it to $Q$ in the next time step.

(ii) Schedule every job in $S$ at rate 1. If a job $j$ has been processed for $\rho_j(1-\alpha)/\alpha$ time steps while being in $S$, remove it from $Q$ and add it to $E$ in the next timestep.

(iii) If a job is completed, remove it from $E$ or $Q$.

For a fixed instance and the algorithm's schedule, we denote by $E_t$, $S_t$ and $Q_t$ the sets $E$, $S$ and $Q$ at time $t$. Note that $Q_t = U_t \setminus E_t$, where $U_t$ denotes the set of unfinished jobs at time $t$.

In the same way as in Algorithm 1, the algorithm gives preferential execution to jobs once they emit their signal (cf. Step (ii)). The queue $Q$ contains all jobs that currently receive preferential execution $(S \cap Q)$ or already emitted and still need preferential execution $(Q \setminus S)$. Since we schedule on $m$ machines, at most $m$ jobs can receive preferential execution at the same time. Hence, the set $S$ contains all jobs that currently receive preferential execution. The set $E$ contains all jobs that have not yet emitted their signal and all jobs that emitted their signal but did not finish during their preferential execution. These jobs are processed in a Round-Robin manner on all machines that are currently not used for preferential execution (cf. Step (i)). In contrast to the single machine setting, the preferential execution of jobs does not necessarily mean that no jobs in $E$ are executed at the same time: If $|S| < m$, then $m - |S|$ machines are not used for the preferential execution of the jobs in $S$ and can be used for Round-Robin execution of the jobs in $E$ instead (cf. the rates in Step (i)).

### E.1 Consistency

The goal of this section is to show the following lemma, which proves the consistency bound of Theorem E.1.

**Lemma E.2.** *If all signals are emitted correctly,* $\text{ALG} \leq (1+\alpha) \cdot \text{OPT}$.

To this end, fix an instance with job lengths $p_1, \ldots, p_n$ and fix the algorithm's schedule assuming that all signals are emitted correctly, i.e., $\beta_j = \alpha$ for all $j \in [n]$.

Define for every job $\ell$ the (possibly empty) set $\mu(\ell) = \{j \in [n] \mid \exists k \in \mathbb{N}_{\geq 1} : j = \ell + k \cdot m\} = \{\ell + m, \ell + 2m, \ldots\}$. Note that in the optimal *shortest processing time first (SPT) schedule*, the set $\mu(\ell)$ contains the jobs that are executed after job $\ell$ on the same machine as job $\ell$. Hence, the optimality of SPT implies

$$\text{OPT} = \sum_{j \in J} \left( p_j + \sum_{\ell \in J : j \in \mu(\ell)} p_\ell \right). \tag{17}$$

For every time $t$ and every job $j \in J$, we define the following values:

- $s_{jt} = \mathbb{1}[j \in U_t \setminus S_t]$, i.e., $s_{jt}$ indicates whether job $j$ does *not* receive preferential execution at time $t$,

- $d_{jt} = \mathbb{1}[j \in U_t \setminus S_t \wedge \nexists \ell \in S_t : j \in \mu(\ell)]$, which indicates whether neither $j$ nor a job that is scheduled before $j$ on the same machine as $j$ in SPT receives preferential execution at $t$, and

- $\Delta_t = \min\{|U_t \setminus S_t|, m\} - \min\{|U_t \setminus S_t|, m - |S_t|\}$, which is the number of machines that are used for preferential executions at time $t$.

Let $T := \max_{j \in [n]} C_j$ denote the time horizon of the algorithm's schedule. We write $d_j = \int_0^T d_{jt} \, dt$ and $s_j = \int_0^T s_{jt} \, dt$. Using this notation, we start by giving a first bound on the algorithms objective value. To this end, we characterize the job completion times as follows.

**Lemma E.3.** *For every job $j$, it holds that* $C_j \leq d_j + (1-\alpha)(p_j + \sum_{\ell \in J : j \in \mu(\ell)} p_\ell)$.

*Proof.* Let $t' \leq T$ be the latest point in time before $j$ enters the set $S$, i.e., before $j$ receives preferential execution for the first time. Since we assume accurate signals, $j$ will complete during its preferential execution within $(1-\alpha)p_j$ time units. Thus, $C_j = t' + (1-\alpha)p_j$.

At any point in time $t \leq t'$, either $d_{jt} = 1$ or some job $\ell \in S_t$ with $j \in \mu(\ell)$ is receiving preferential execution (using the definition of $d_{jt}$ and that $j \notin S_t$ by choice of $t'$). The maximum amount of time during which jobs $\ell$ with $j \in \mu(\ell)$ receive preferential execution is $(1-\alpha) \sum_{\ell \in J : j \in \mu(\ell)} p_\ell$, again

using that in the case of accurate signals each such job $\ell$ receives preferential execution for $(1 - \alpha)p_\ell$ time. Hence, $t' \leq d_j + (1 - \alpha) \sum_{\ell \in J : j \in \mu(\ell)} p_\ell$ and, therefore,

$$C_j = t' + (1 - \alpha)p_j \leq d_j + (1 - \alpha)(p_j + \sum_{\ell \in J : j \in \mu(\ell)} p_\ell).$$

$\square$

This upper bound on the job completion times and the characterization of OPT in (17) immediately give

$$\mathrm{ALG} = \sum_{j \in [n]} C_j \leq \sum_{j \in [n]} d_j + (1 - \alpha) \sum_{j \in J} \left( p_j + \sum_{\ell \in J : j \in \mu(\ell)} p_\ell \right)$$

$$\leq \sum_{j \in [n]} d_j + (1 - \alpha)\mathrm{OPT}.$$

Hence, it remains to bound $\sum_{j \in [n]} d_j$ by $2\alpha \cdot \mathrm{OPT}$ to prove the consistency of $(1 + \alpha)$. To do so, we continue with the following auxiliary lemma, which allows us to replace $\sum_{j \in [n]} d_j$ with $\sum_{j \in [n]} s_j - \int_0^T \Delta_t \, dt$ in the inequality above.

**Lemma E.4.** *At any time $t \leq T$ it holds that $\sum_{j \in [n]} s_{jt} - d_{jt} \geq \Delta_t$.*

*Proof.* To prove the statement, we first make the following two observations

1. Every job $j \in U_t \setminus S_t$ contributes a value of exactly one to the left side of the inequality only if there is a job $\ell \in S_t$ s.t. $j \in \mu(\ell)$. Hence,

$$\sum_{j \in [n]} s_{jt} - d_{jt} \geq |\{ j \in U_t \setminus S_t \mid \exists \ell \in S_t : j \in \mu(\ell) \}|.$$

2. The algorithm completes the jobs in order of their indices. To see this, fix two jobs $j, i$ with $p_j < p_i$. By definition of the algorithm, the two jobs are processed with the same rate until $j$ emits its signal and is removed from $E$ and added to $Q$. Since $j$ is added to $Q$ before $i$, the job $j$ will enter the set $S$ earlier than job $i$. Hence, the preferential execution of $j$ starts earlier than the preferential execution of job $i$ and, using $p_j < p_i$, is also shorter. Since $j$ completes during its preferential execution, this implies $C_j < C_i$. For jobs $j, i$ with $p_i = p_j$, we can argue in the same way that they enter $Q$ at the same time. By using the same tiebreaking rule in the queue $Q$ and in the job index order, we can achieve that the algorithm finishes the jobs in order of their indices.

Since the jobs complete in order of their index by the second observation above, we can assume that $S_t = \{k, k + 1, \ldots, k + |S_t| - 1\}$ and $U_t \setminus S_t = \{k + |S_t|, \ldots, n\}$ for some $k \in [n]$. Therefore, for every job $j \in U_t \setminus S_t$ with $k + m \leq j \leq \min\{n, k + m + |S_t| - 1\}$ it holds that $j \in \mu(j - m)$, and $j - m \in S_t$. Note that if $k + m > n$, there are no such jobs. Thus,

$$|\{ j \in U_t \setminus S_t \mid \exists \ell \in S_t : j \in \mu(\ell) \}|$$
$$= \min\{n, k + m + |S_t| - 1\} - \min\{n + 1, k + m\} + 1$$
$$= \min\{|U_t| - 1, m + |S_t| - 1\} - \min\{|U_t|, m\} + 1$$
$$= (\min\{|U_t \setminus S_t|, m\} + |S_t|) - (\min\{|U_t \setminus S_t|, m - |S_t|\} + |S_t|) = \Delta_t.$$

Using the first observation from the beginning of the proof, we can conclude with

$$\sum_{j \in [n]} s_{jt} - d_{jt} \geq |\{ j \in U_t \setminus S_t \mid \exists \ell \in S_t : j \in \mu(\ell) \}| \geq \Delta_t.$$

$\square$

As argued above, Lemmas E.3 and E.4 and (17) together imply,

$$\text{ALG} \le \sum_{j \in [n]} s_j - \int_0^T \Delta_t \, \mathrm{d}t + (1 - \alpha)\text{OPT}.$$

To conclude the proof of the consistency case ( Lemma E.2), it remains to upper bound $\sum_{j \in [n]} s_j - \int_0^T \Delta_t \, \mathrm{d}t$ by $2\alpha \cdot \text{OPT}$, which we do with the following lemma.

**Lemma E.5.** *Let* $\text{RR}(\{\alpha p_j\}_{j \in [n]})$ *denote the total completion time of the Round-Robin-produced schedule for the instance characterized by the processing times* $\{\alpha p_1, \dots, \alpha p_n\}$. *Then,*

- $\text{RR}(\{\alpha p_j\}_{j \in [n]}) \le 2 \cdot \text{OPT}(\{\alpha p_j\}_{j \in [n]}) = 2\alpha \cdot \text{OPT}$, *and*

- $\text{RR}(\{\alpha p_j\}_{j \in [n]}) = \sum_{j \in [n]} s_j - \int_0^T \Delta_t \, \mathrm{d}t$.

*Proof.* The first claim, $\text{RR}(\{\alpha p_j\}_{j \in [n]}) \le 2 \cdot \text{OPT}(\{\alpha p_j\}_{j \in [n]}) = 2\alpha \cdot \text{OPT}$, follows because Round-Robin is 2-competitive [MPT94] and because scaling all processing times by the same factor does not change the structure of an optimal schedule, hence only scales the job completion times.

For the second claim, we consider the Round-Robin schedule $\mathcal{S}_0$ for the instance $\{\alpha p_j\}_{j \in [n]}$ and the schedule $\mathcal{S}_T$, which is the schedule computed by the algorithm *without* the preferential execution. That is, we take the algorithms schedule for the instance $\{p_j\}_{j \in [n]}$ and replace all preferential executions with idle time. The latter is equivalent to, at any time $t$, changing the rates of jobs $j \in S_t$ from $R_j^t = 1$ to $R_j^t = 0$. Note that $\mathcal{S}_T$ is a valid schedule for instance $\{\alpha p_j\}_{j \in [n]}$ since the accurate signals imply that $\mathcal{S}_T$ processes every job $j$ for exactly $\alpha p_j$ time units. The objective value of schedule $\mathcal{S}_T$ for the instance $\{\alpha p_j\}_{j \in [n]}$ is exactly $\sum_{j \in [n]} C_j(\mathcal{S}_T) = \sum_{j \in [n]} s_j$, as $s_j$ is the point in time at which the algorithm completes the first $\alpha$-fraction of job $j$. Here, we use $C_j(\mathcal{S})$ to denote the completion time of job $j$ in schedule $\mathcal{S}$ for instance $\{\alpha p_j\}_{j \in [n]}$. On the other hand, the objective value of $\mathcal{S}_0$ for the instance $\{\alpha p_j\}_{j \in [n]}$ is exactly $\sum_{j \in [n]} C_j(\mathcal{S}_0) = \text{RR}(\{\alpha p_j\}_{j \in [n]})$. Our strategy for proving the second claim of the lemma is to show that

$$\sum_{j \in [n]} C_j(\mathcal{S}_T) - \sum_{j \in [n]} C_j(\mathcal{S}_0) = \int_0^T \Delta_t \, \mathrm{d}t \,, \tag{18}$$

which then implies the second claim of the lemma.

To show that (18) indeed holds, we consider the series of schedules $\mathcal{S}_t$, $0 \le t \le T$, for the instance $\{\alpha p_j\}_{j \in [n]}$, where each $\mathcal{S}_t$ is defined as follows:

(i) Define $U_{t'}(\mathcal{S}_t)$ to be the set of unfinished jobs in schedule $\mathcal{S}_t$ at time $t'$. For every time $t' \le t$ and every $j \in U_{t'}(\mathcal{S}_t)$, define $R_j^{t'} = q_{t'}$. Recall that $q_{t'}$ is the rate that the jobs in $U_{t'} \setminus S_{t'}$ receive at time $t'$ in the algorithms schedule for the instance $\{p_j\}_{j \in [n]}$ and, thus, also the rate that the job $j$ receives at time $t'$ in schedule $\mathcal{S}_T$.

Note that this definition implies $U_{t'}(\mathcal{S}_t) = U_{t'} \setminus Q_{t'}$ for all $t' \le t$, as $\mathcal{S}_t$ operates on instance $\{\alpha p_j\}_{j \in [n]}$ using the same rates as the algorithm uses on the full instance, which means that $U_{t'} \setminus Q_{t'}$ contains exactly all jobs that have not yet completed an $\alpha$-fraction of their processing time.

(ii) For every time $t' > t$ and every job $j \in U_{t'}(\mathcal{S}_t)$, it holds that $R_j^{t'}(\mathcal{S}_t) = \min\{|U_{t'}(\mathcal{S}_t)|, m\}/|U_{t'}(\mathcal{S}_t)|$. That is, the sub schedule for the sub interval $(t, T]$ uses exactly the Round-Robin rates.

Intuitively, for any $0 \le t \le T$, the schedule $\mathcal{S}_t$ uses the same rates as $\mathcal{S}_T$ during $[0, t]$ and uses Round-Robin during $(t, T]$. Let $\mathcal{T} \subseteq [0, T]$ denote the set of points in time with $0, T \in \mathcal{T}$ that additionally contains all times $t$ in $(0, T)$ at which the rates in any of the schedules $\mathcal{S}_{t'}$ change. For all $t \in \mathcal{T} \setminus \{0\}$, let $t^- := \max\{t' \in \mathcal{T} \mid t' < t\}$. We conclude the proof by showing that

$$\sum_{j \in [n]} C_j(\mathcal{S}_t) - \sum_{j \in [n]} C_j(\mathcal{S}_{t^-}) = \int_{t^-}^t \Delta_{t'} \, \mathrm{d}t' \tag{19}$$

holds for all $t \in \mathcal{T} \setminus 0$, which the implies (18) and, thus, the lemma.

To this end, consider an arbitrary $t \in \mathcal{T} \setminus \{0\}$ and the corresponding $t^-$. In the following, we use $e_j^{t'}(\mathcal{S})$ to denote the elapsed time of $j$ at $t'$ in schedule $\mathcal{S}$. By definition (see (i)) the schedules $\mathcal{S}_t$ and $\mathcal{S}_{t^-}$ use the exact same rates during $[0, t^-]$. Hence, all jobs have the same elapsed time at time $t^-$ in both schedules, i.e., $e_j^{t^-}(\mathcal{S}_t) = e_j^{t^-}(\mathcal{S}_{t^-})$ for all $j \in [n]$.

Next, consider the interval $(t^-, t]$. By definition of $\mathcal{T}$, both schedules do not change their respective processing rates during $(t^-, t)$. Using (i), this also means that both schedules process the jobs in $U_{t^-} \setminus S_{t^-}$ during the entire interval. Thus, schedule $\mathcal{S}_{t^-}$ processes exactly the jobs $j \in U_{t^-} \setminus S_{t^-}$ with rate

$$\frac{\min\{|U_{t^-} \setminus S_{t^-}|, m\}}{|U_{t^-} \setminus S_{t^-}|},$$

whereas $\mathcal{S}_t$ processes exactly the jobs $j \in U_{t^-} \setminus S_{t^-}$ with rate

$$\frac{\min\{|U_{t^-} \setminus S_{t^-}|, m - |S_{t^-}|\}}{|U_{t^-} \setminus S_{t^-}|} = \frac{\min\{|U_{t^-} \setminus S_{t^-}|, m\}}{|U_{t^-} \setminus S_{t^-}|} - \frac{\Delta_t}{|U_{t^-} \setminus S_{t^-}|}.$$

Thus, $e_j^t(\mathcal{S}_t) = e_j^t(\mathcal{S}_{t^-}) - \int_{t^-}^t \frac{\Delta_{t'}}{|U_{t^-} \setminus S_{t^-}|} \, \mathrm{d}t'$ for all $j \in U_{t^-} \setminus S_{t^-}$.

Finally, after $t$, both schedules $\mathcal{S}_t$ and $\mathcal{S}_{t^-}$ execute round-robin with the only difference being that the elapsed time in $\mathcal{S}_{t^-}$ are already farther advanced than the elapsed times in $\mathcal{S}_t$. Since the difference in the elapsed times is the same for all jobs in $U_{t^-} \setminus S_{t^-}$, the round-robin schedule in $\mathcal{S}_t$ after $t$ is equivalent to the round-robin schedule in $\mathcal{S}_{t^-}$ after $t - \int_{t^-}^t \frac{\Delta_{t'}}{|U_{t^-} \setminus S_{t^-}|} \, \mathrm{d}t'$. This means that the completion times of jobs $j \in U_{t^-} \setminus S_{t^-}$ in schedule $\mathcal{S}_t$ are the same as in $\mathcal{S}_{t^-}$ but delayed by $\int_{t^-}^t \frac{\Delta_{t'}}{|U_{t^-} \setminus S_{t^-}|} \, \mathrm{d}t'$. We can conclude with

$$\sum_{j \in [n]} C_j(\mathcal{S}_t) = \sum_{j \in [n] \setminus (U_{t^-} \setminus S_{t^-})} C_j(\mathcal{S}_{t^-}) + \sum_{j \in U_{t^-} \setminus S_{t^-}} \left( C_j(\mathcal{S}_{t^-}) + \int_{t^-}^t \frac{\Delta_{t'}}{|U_{t^-} \setminus S_{t^-}|} \, \mathrm{d}t' \right)$$

$$= \left( \sum_{j \in [n]} C_j(\mathcal{S}_{t^-}) \right) + \int_{t^-}^t \Delta_{t'} \, \mathrm{d}t',$$

which implies (19) and, thus, the lemma. $\qquad \square$

## E.2 Robustness

In this section, we prove the robustness bound of $\mathrm{ALG} \leq (1 + 1/\alpha) \cdot \mathrm{OPT}$. To do so, we heavily rely on the following lower bound on the optimal objective value, which is a mixed bound of the standard volume lower bound and the fast single machine lower bound.

**Lemma E.6** (Mixed lower bound; Lemma 9 in [BBEM12]). *For minimizing the total completion time of $n$ jobs with processing times $p_1 \leq \ldots \leq p_n$ on $m$ parallel identical machines, given a partition $p_j^{(1)} + p_j^{(2)} \leq p_j$ for every job $j$, it holds that*

$$\frac{1}{m} \sum_{j=1}^n \sum_{k=1}^j p_k^{(1)} + \sum_{j=1}^n p_j^{(2)} \leq \mathrm{OPT}.$$

Before we prove the robustness bound, recall that $T$ is the time horizon of the algorithm's schedule and define $T'$ as the earliest point in time with $|U_{T'}| \leq m$. Starting from $T'$, the number of alive jobs in the algorithm's schedule is at most the number of machines, which implies that the algorithm processes all remaining jobs with the maximum rate of one until they complete.

**Lemma E.7.** *For every $\alpha \in (0, 1]$ it holds that $\mathrm{ALG} \leq (1 + 1/\alpha) \cdot \mathrm{OPT}$.*

*Proof.* In order to later apply the lower bound on OPT of Lemma E.6, we first split the processing times $p_j$ of the jobs $j \in [n]$:

- Let $p_j^S = \int_0^T \mathbb{1}[j \in S_{t'}] \, dt'$ denote the amount of processing that $j$ receives at times $t$ with $j \in S_t$. That is, $p_j^S$ is the amount of preferential execution that $j$ receives.

- Let $p_j^F = \int_{T'}^T \mathbb{1}[j \in E_{t'}] \, dt'$ denote the amount of processing that $j$ receives after $T'$ while being in $E$. Note that after $T'$, the job $j$ will be executed with the full rate of one even if it is in $E$, as the number of alive jobs is at most the number of machines.

- Finally, let $p_j^R = p_j - p_j^F - p_j^S$ denote the amount of processing that $j$ receives with a rate strictly less than one.

Next, recall that $q_t$ is the rate at which all jobs in $E_t$ are executed at time $t$. Observe that $q_t = (m - |S_t|)/|E_t|$ holds for all $t \le T'$ by definition of the algorithm and since $|U_t| > m$ by choice of $T'$. Therefore,

$$\frac{1}{|E_t|} = \frac{q_t}{m} + \frac{|S_t|}{m \cdot |E_t|} \tag{20}$$

for all $t < T'$. This allows us to derive a first upper bound on ALG:

$$\begin{aligned}
\text{ALG} &= \int_0^T |U_t| \, dt = \int_0^T |E_t| \, dt + \int_0^T |U_t \setminus E_t| \, dt \\
&= \int_0^{T'} |E_t|^2 \cdot \frac{1}{|E_t|} \, dt + \int_{t=T'}^T |E_t| \, dt + \int_0^T |U_t \setminus E_t| \, dt \\
&= \int_0^{T'} |E_t|^2 \cdot \frac{q_t}{m} \, dt + \int_0^{T'} \frac{|S_t| \cdot |E_t|}{m} \, dt + \sum_{j=1}^n p_j^F + \int_0^T |U_t \setminus E_t| \, dt \\
&\le \frac{2}{m} \int_0^{T'} \left( \sum_{j \in E_t} \sum_{\substack{k \in E_t \\ k \le j}} q_t \right) dt + \int_0^{T'} \frac{|S_t| \cdot |E_t|}{m} \, dt + \sum_{j=1}^n p_j^F + \int_0^T |U_t \setminus E_t| \, dt \\
&\le \frac{2}{m} \sum_{j=1}^n \sum_{k=1}^j \int_0^{T'} q_t \cdot \mathbb{1}[k \in E_t] \, dt + \int_0^{T'} \frac{|S_t| \cdot |E_t|}{m} \, dt + \sum_{j=1}^n p_j^F + \int_0^T |U_t \setminus E_t| \, dt \\
&= \frac{2}{m} \sum_{j=1}^n \sum_{k=1}^j p_k^R + \int_0^{T'} \frac{|S_t| \cdot |E_t|}{m} \, dt + \sum_{j=1}^n p_j^F + \int_0^T |U_t \setminus E_t| \, dt \tag{21}
\end{aligned}$$

To further upper bound (21), we first focus on upper bounding the sum of the two integrals in (21), which we rewrite as follows:

$$\begin{aligned}
&\int_0^{T'} \frac{|S_t| \cdot |E_t|}{m} \, dt + \int_0^T |U_t \setminus E_t| \, dt \\
&= \int_0^{T'} \frac{|S_t| \cdot |E_t|}{m} \, dt + \int_0^T \mathbb{1}[S_t \subsetneq U_t \setminus E_t] \cdot |U_t \setminus E_t| \, dt + \int_0^T \mathbb{1}[S_t = U_t \setminus E_t] \cdot |U_t \setminus E_t| \, dt.
\end{aligned}$$

Note that at any time when $S_t \subsetneq U_t \setminus E_t$ it must hold $|S_t| = m$. To see this, recall that $S_t$ is defined to contain the first up-to $m$ elements in $Q_t = U_t \setminus E_T$ (see the definition of the algorithm). Hence, if $S_t$ does not contain all elements of $Q_t$, then it must contain $m$ elements. Thus, the above is equal to

$$\int_0^{T'} \frac{|S_t| \cdot |E_t|}{m} \, dt + \int_0^T \mathbb{1}[S_t \subsetneq U_t \setminus E_t] \cdot |U_t \setminus E_t| \cdot \frac{|S_t|}{m} \, dt + \int_0^T \mathbb{1}[S_t = U_t \setminus E_t] \cdot |U_t \setminus E_t| \, dt$$

$$\leq \frac{1}{m} \int_0^{T'} |S_t| \cdot |E_t| \, dt + \frac{1}{m} \int_0^T \mathbb{1}[S_t \subsetneq U_t \setminus E_t] \cdot |U_t \setminus E_t| \cdot |S_t| \, dt + \int_0^T |S_t| \, dt$$

$$\leq \frac{1}{m} \int_0^{T'} |S_t| \cdot |E_t| \, dt + \frac{1}{m} \int_0^T |U_t \setminus E_t| \cdot |S_t| \, dt + \int_0^T |S_t| \, dt$$

$$\leq \frac{1}{m} \int_0^T |S_t| \cdot |U_t| \, dt + \int_0^T |S_t| \, dt$$

$$\leq \frac{1}{m} \sum_{j=1}^n \sum_{k=1}^j \int_0^T \mathbb{1}[j \in S_t \wedge k \in U_t] + \mathbb{1}[k \in S_t \wedge j \in U_t] \, dt + \sum_{j=1}^n p_j^S. \tag{22}$$

For a job $j$, let $t_j$ be the time it enters set $S$. If there is another job $j$ that is unfinished at time $t_j$, then we can observe $e_j(t_j) \leq e_k(t_j)$: In case $k$ has not yet entered $S$ at $t_j$, both jobs have been processed with the same rates during $[0, t_j)$, which implies $e_j(t_j) = e_k(t_j)$. On the other hand, if $k$ entered $S$ earlier, then $R_k^{t'} \geq R_j^{t'}$ for all $t' < t_j$ as $k$ either has the same rate as $j$ (if $k \in U_{t'} \setminus S_{t'}$) or receives preferential execution and has a larger rate as $j$ (if $k \in S_{t'}$). This implies $e_j(t_j) \leq e_k(t_j)$. From $e_j(t_j) \leq e_k(t_j)$, we get $\rho_j := e_j(t_j) \leq e_k(t_j) \leq p_k$. By definition of the algorithm, $j$ receives preferential execution for $\frac{1-\alpha}{\alpha} \rho_j \leq \frac{1-\alpha}{\alpha} p_k$ time units. Thus, $j$ is part of set $S$ for at most $\frac{1-\alpha}{\alpha} p_k$ time units. This implies

$$\int_0^T \mathbb{1}[j \in S_t \wedge k \in U_t] \, dt \leq \frac{1-\alpha}{\alpha} \cdot p_k.$$

Furthermore, since $p_k^S = \int_0^T \mathbb{1}[j \in S_t] \, dt$ holds by definition of $p_k^S$, we also get

$$\int_0^T \mathbb{1}[k \in S_t \wedge j \in U_t] \, dt \leq p_k^S.$$

Plugging these two inequalities into (22) yields

$$\int_0^{T'} \frac{|S_t| \cdot |E_t|}{m} \, dt + \int_0^T |U_t \setminus E_t| \, dt$$

$$\leq \frac{1}{m} \sum_{j=1}^n \sum_{k=1}^j \int_0^T \mathbb{1}[j \in S_t \wedge k \in U_t] + \mathbb{1}[k \in S_t \wedge j \in U_t] \, dt + \sum_{j=1}^n p_j^S$$

$$\leq \frac{1-\alpha}{\alpha} \frac{1}{m} \sum_{j=1}^n \sum_{k=1}^j p_k + \frac{1}{m} \sum_{j=1}^n \sum_{k=1}^j p_k^S + \sum_{j=1}^n p_j^S.$$

To conclude the proof, we can plug this inequality into (21):

$$\text{ALG} \leq \frac{2}{m} \sum_{j=1}^n \sum_{k=1}^j p_k^R + \int_0^{T'} \frac{|S_t| \cdot |E_t|}{m} \, dt + \sum_{j=1}^n p_j^F + \int_0^T |U_t \setminus E_t| \, dt$$

$$\leq \frac{1-\alpha}{\alpha} \frac{1}{m} \sum_{j=1}^n \sum_{k=1}^j p_k + \frac{2}{m} \sum_{j=1}^n \sum_{k=1}^j p_k^R + \sum_{j=1}^n p_j^F + \frac{1}{m} \sum_{j=1}^n \sum_{k=1}^j p_k^S + \sum_{j=1}^n p_j^S$$

$$\leq \left( \frac{1-\alpha}{\alpha} \right) \text{OPT} + \frac{1}{m} \sum_{j=1}^n \sum_{k=1}^j \left( p_k^R + p_k^S \right) + \sum_{j=1}^n p_j^F + \frac{1}{m} \sum_{j=1}^n \sum_{k=1}^j p_k^R + \sum_{j=1}^n p_j^S$$

$$\leq \left( \frac{1-\alpha}{\alpha} \right) \text{OPT} + \text{OPT} + \text{OPT} \leq \left( 1 + \frac{1}{\alpha} \right) \text{OPT}.$$

Here, the second inequality uses Lemma E.6 with the partition $p_j^{(1)} = p_j$ and $p_j^{(2)} = 0$. The third inequality we use Lemma E.6 once with the partition $p_j^{(1)} + p_j(2) = (p_j^R + p_j^S) + p_j^F = p_j$ and once with the partition $p_j^{(1)} + p_j(2) = p_j^R + p_j^S \leq p_j$. $\qquad \square$

