# OpenReview forum: "Non-Clairvoyant Scheduling with Progress Bars"
_NeurIPS.cc/2025/Conference — NeurIPS 2025 spotlight_

### Official Review · Reviewer_Ur9h · 2025-06-22

**Clarity:** 2
**Significance:** 3
**Originality:** 3
**Rating:** 4
**Confidence:** 3

**Summary:**

This paper considered several settings under non-clairvoyant scheduling, namely single signal progress bar, untrusted and stochastic. In single signal progress bar, the paper introduced several algorithms and proved the consistency upper bound of these algorithm. The paper further discussed the robustness and smoothness of these algorithms, and proved that the proposed Algorithm 1 achieves a tradeoff between smoothness and robustness. The paper further proposed a lower bound of robustness.

The paper then generalized the results to untrusted setting and stochastic setting. In both setting, algorithms are introduced with corresponding consistency upper bound.

**Questions:**

My questions are listed below:

1. What is the major technical difficulties to achieve the performance guarantees?

2. For the problem described in Section 3, is there a lower bound for the total completion time?

**Ethical Concerns:**

["NO or VERY MINOR ethics concerns only"]

**Final Justification:**

All my concerns are addressed by the authors.

**Limitations:**

Limitations are not explicitly discussed in this paper. See questions and weakness for potential limitation.

**Quality:**

3

**Strengths And Weaknesses:**

Strengths:

1. This paper proposed a new problem setting, offline scheduling with progress bar feedback, which is theoretically interesting and practically relevant.

2. Under single-signal setup, the paper discussed several strategies achieving optimal consistency and tradeoffs between robustness and smoothness. These results are new to me.

3. The paper also considered stochastic progress bar, where the signals are emitted according to Poisson process. This setting is also interesting and sounds practical relevant. Meanwhile, Algorithm 3, designed for this setting, with theoretical guarantee, is also new to me.

Weakness:

1. In the proof of Theorem 2.5, line 586, the claim "the algorithm processes $I_1$ and $I_2$ equivalently until time t" requires more clarification. Since $p_j$ are different in $I_1$ and $I_2$, the time of emitting signals might be different, which might results to a different execution routine.

2. Algorithm 2 looks like a direct generalization of the results in [EKMM24].

3. The presentation of this paper is not clear enough. For example, the definition of robustness and brittleness are neither presented in the paper nor referred to previous works.

---

> ### Author Rebuttal · Authors · 2025-07-30
>
> We thank the reviewer for their interest in our work and for reading and assessing our paper carefully. In the following, we comment on weaknesses and questions raised by the reviewer.
>
> **Clarity of the presentation:** It is true that the present version of the paper provides very succinct inline definitions of consistency, robustness, and smoothness (cf. Lines 34-37), as well as references to the literature that introduced these terms (cf. Line 32). The same holds for the term brittleness, which we only describe informally (cf. Lines 178-179) and with references to relevant related work (cf. Line 177).  For the sake of this paper, the statement of Proposition A.3 in the appendix can be understood as a definition of a brittle algorithm: A small perturbation in the signals leads to a competitive ratio that does not improve upon the lower bound for non-clairvoyant algorithms without access to untrusted signals.  We are happy to add formal definitions of all these terms in the final version of our paper.
>
> **Main technical challenges:** A key difference between all models considered in this paper and previous works on learning-augmented algorithms for non-clairvoyant scheduling is that algorithms in our models start without additional information on the instance. In contrast to these previous works, even perfect signals are not sufficient for designing an 1-competitive algorithm, because the algorithm will already make mistakes before it receives additional information. The overarching challenge across all our results is to design algorithms that do not make too many mistakes before more information is revealed. In the adversarial settings, we achieve this by carefully transitioning between different well-established algorithms for sum of completion time scheduling, and analyze our algorithms with a carefully parameterized case distinction. In the stochastic setting, we achieve this with an algorithm that uses ideas from multi-armed bandits (e.g., Explore-Then-Commit) and tailors them to the scheduling problem at hand (e.g., the Round-Robin phase and the repeated commit phases). The theoretical results for this algorithm require a non-trivial analysis that integrates scheduling aspects (e.g., the analysis of job delays) into a stochastic analysis.
>
> **Proof of Theorem 2.5:** Thank you for pointing this out, we will add a clarification in the final version. In the proof, we construct the two instances $I_1$ and $I_2$. For instance $I_1$, we assume that the signals are emitted correctly in order to enforce the inequality (3) for every deterministic $(1+\alpha)$-consistent algorithm. Since our goal is to show a lower bound on the robustness for every deterministic $(1+\alpha)$-consistent algorithm, we do not need to define $I_2$ with correctly emitted signals, and instead use adversarial signals that can be wrong. This allows us to just define the adversarial signals of instance $I_2$ that are emitted during $[0,t]$ to be exactly as in instance $I_1$. Hence, a deterministic algorithm is not able to distinguish $I_1$ and $I_2$ during $[0,t]$, and will process both instances in the same way. We will clarify that the equivalent behavior during $[0,t]$ on both instances is a consequence of the adversarial signals on instance $I_2$.
>
> **Lower bounds for scheduling with an untrusted progress bar (Section 3):** Lower bounds for the special case of a single untrusted signal (Section 2, Theorem 2.5) naturally translate to the more general setting of Section 3. Besides this, [EKMM24] gives a lower bound of $\Omega(ng)$ on the regret term for combining $g$ algorithms (Theorem 32 in the arxiv version). In their lower bound, each algorithm is represented by a predicted job size vector and one of  these vectors matches the actual processing times. This lower bound instance should naturally translate to the untrusted progress bar setting, where we can just replace the $g$ different predicted sizes for a job with $g$ signals that are consistent with these predicted job sizes. Since this only reduces the amount of information the algorithm has access to, the lower bound should still hold. We will add a comment on this lower bound in the final version of the paper. Apart from this, we are not aware of any lower bounds.

---

> > ### Comment · Reviewer_Ur9h · 2025-08-01
> >
> > I thank the authors for the response and would like to keep my positive rating

---

### Official Review · Reviewer_AbAQ · 2025-07-01

**Clarity:** 4
**Significance:** 3
**Originality:** 3
**Rating:** 4
**Confidence:** 4

**Summary:**

The paper considers the single machine non-clairvoyant scheduling to minimize the total completion time with progress bars. All jobs are available at time 0 with an unknown job size. As a job is being processed, estimates of the job's progress are received by the scheduler as progress bars, allowing for informed decisions. The progress bars may be inaccurate, meaning that they may misreport the actual fractions of the processed job. Thus, this work falls within the field of algorithms with predictions, where the algorithm is expected to use predictions so that 1) the performance is near-optimal when the predictions are perfect (consistency) and 2) the performance is bounded under arbitrarily bad predictions (robustness). The distinction of this work is that the progress bars (i.e., the predictions) are received in the process of scheduling, contrasting the existing works where predictions are static and received upfront. The authors considered various settings of the progress bars and gave solutions for each: single signal progress bars, progress bars with multiple levels, and stochastic progress bars. For each setting, a consistent and robust algorithm is proposed. Experiments were conducted to validate the proposed algorithms.

**Questions:**

1. Regarding Line 204, the claimed contribution of improving upon the state-of-the-art seems a bit oversold. The parameter alpha must be chosen arbitrarily for the results to hold. Can you please justify this point?

2. How are computable delays used in the proof of Theorem 3.1? Corollary 3.2 requires the algorithms to have computable delays, which were not shown. This may affect the correctness of the corollary. Can authors explain this point? The validity of the correctness may affect my evaluation. For example, if the result is not correctly proved, my evaluation may be lowered.

3. Line 341. It is not clear that the delayed predictions have performance which is highly sensitive to errors. It seems that the performance deviation is small. Can you please explain this?

**Ethical Concerns:**

["NO or VERY MINOR ethics concerns only"]

**Quality:**

3

**Strengths And Weaknesses:**

Strengths:

1. Scheduling with predictions continuously arrived at the scheduler is a novel setting. This setting is practical so algorithmic advances along this dimension are valuable.

2. The problem of scheduling with progress bars was thoroughly studied under adverbial and stochastic settings. The results are rich.

3. The paper makes meaningful connections to other fields, including the classic non-clairvoyant scheduling with predictions and multi-armed bandits.


Weaknesses:

There are two major weaknesses and a few minor ones identified. The first major one is that some statements feel like over-selling the contributions, and the second is that elaboration and clarity are needed around the computable delays.


Overselling contributions:

1. Line 204: The contribution of improving upon state-of-the-art seems a bit oversold. The parameter alpha needs to be able to choose freely for the result to hold. It is not clear if the parameter alpha can be chosen freely. It is at least not explicitly stated so.

2. Line 267 claims that a near-perfect consistency and robustness could be simultaneously achievable, but it is a bit oversold as it requires specific problem instances, e.g., max_i p_i = o(n^{-5/3} OPT).


Computable delays:

1. The need for computable delays is not explicitly in the proof of Theorem 3.1. What would happen to which part of the proof if the algorithms do not have computable delays?

2. Corollary 3.2 requires algorithms (A^{(h)})_h to have computable delays to use Theorem 3.1. This was not shown, which may affect the correctness of the corollary. Are mutual delays of these algorithms functions in progress levels alpha and beta?


Minor issues:

1. Line 241: typo OPT = \Omega not \omega.

2. Line 612: formula 2mn p_{max} is missing.

3. Line 617: all with variance factor m p^2_{\max}.


Despite the issues identified, I have an overall positive feeling about the paper. The contributions are sufficient for NeurIPS, conditional on that the Corollary 3.2 is correctly proved, where I have a few questions in the Question section.

---

> ### Author Rebuttal · Authors · 2025-07-30
>
> We thank the reviewer for their time, effort, and feedback on our submission. We address their questions and comments below.
>
> **Question 1.** In the context of scheduling with progress bars in Section 2, the parameter $\alpha$ is indeed part of the problem definition and cannot be freely chosen. Nonetheless, our results can be applied to the setting of non-clairvoyant scheduling with predicted processing times $(\pi_i)_i$, provided as inputs. Indeed, we can choose any $\alpha \in [0, 1]$ and "simulate" a progress bar that jumps at fraction $\alpha$, i.e., when job $i$ has been processed for a time $\alpha \pi_i$, we consider that the progress bar jumped to level $\alpha$, and we use our algorithm. This ensures the claimed consistency and robustness. More details are provided in Lines 196–203. We will revise the corresponding paragraph to emphasize that, in this setting, $\alpha$ is a free parameter that can be chosen arbitrarily in the algorithm.
>
> **Question 2.** Regarding the role of computable delays: these are not used in the proof of Theorem 3.1, but they are essential for defining Algorithm 2. Specifically, after executing all jobs $(u_k, v_k)_{k=1}^m$ to completion, the algorithm must compute $d^{(h)}(u_k, v_k)$ for each $k$. The assumption of computable delays ensures this computation is possible. We will clarify this point in the paper.
>
> For Corollary 3.2, we combine Round-robin with the algorithm that blindly follows the predictions. Round-robin has computable delays because for any jobs $i$ and $j$, we have $d(i,j) = \min(p_i, p_j)$, which can be computed once both jobs have completed and their processing times are known. For the algorithm that blindly follows the predictions, we show in the proof of Lemma 2.2 that:
> - if $\beta_i p_i < \beta_j p_j$, then $d(i,j) = p_i$
> - otherwise, $d(i,j) = \beta_j p_j$
>
> Thus, $d(i,j)$ depends on $(\beta_i p_i, \beta_j p_j, p_i)$, all of which are known after both jobs are completed, since $\beta_i p_i$ and $\beta_j p_j$ are the times when the jobs emit their signals, and $p_i$ is observed at completion. We agree with the reviewer that this is an important explanation, and we will include it in the main body of the paper.
>
> **Question 3.** Figure 2 shows the empirical competitive ratio of the algorithms (on the y-axis) as a function of the prediction error parameter $\sigma$ (on the x-axis). For delayed predictions, when $\sigma = 0$, the competitive ratio is 1.5. However, even for very small $\sigma > 0$, the ratio quickly degrades to 2, indicating a sharp decline in performance. This illustrates the algorithm’s sensitivity to even minor prediction errors. We agree that the phrase “the performance is highly sensitive to errors” may be ambiguous, and we will revise the text to explain this observation more clearly.
>
> **Minor issues.** We thank the reviewer for bringing the minor issues to our attention. We will correct them.

---

> > ### Comment · Reviewer_AbAQ · 2025-08-06
> > **Still having concerns in relation to Corollary 3.2.**
> >
> > Question 1 is answered well. I have no further concerns.
> >
> > The response to Question 2 confused me and thus raised a red flag regarding the correctness of Corollary 3.2. The definition of computable delays (introduced in Line 223 of the paper) requires that d depends only on the job processing times; the author's response clearly shows that d also depends on betas. To me, this means that blindly following the predictions does not have computable delays. Therefore, I suspect that the proof for Corollary 3.2 has flaws. This also shows that the assumption of most standard scheduling algorithms having computable delays (in Line 225) does not hold; subsequently, the generality of Theorem 3.1 is questionable. My initial evaluation of acceptance is conditional on the correctness of this result. Therefore, I will lower my score but will be happy to raise it back upon clear justifications.
> >
> > Question 3 is answered well. I have no further concerns.
> >
> > Action: I recommend revising the score to 3. Borderline reject.

---

> ### Author Response · Authors · 2025-08-07
>
> We thank the reviewer for engaging in this discussion.
> We agree with the reviewer that the definition of computable delays (line 223) should be revised. The current definition is:
> $$d(i,j) \text{ can be computed knowing only } p_i \text{ and } p_j.$$
>
> However, the definition that Corollary 3.2 uses is slightly different:
> $$ d(i,j) \text{ can be computed once both jobs } i \text{ and } j \text{ have completed.} $$
>
> Theorem 3.1 holds for both definitions, but the second definition is stronger than the first, as it allows the algorithm to use not only the exact processing times $p_i$ and $p_j$, which become known only after job completion in non-clairvoyant settings, but also any additional information observed before completion (such as progress bars, observed predictions), as well as all inputs to the problem (such as static predictions).
>
> Importantly, in non-clairvoyant settings, with or without predictions, the exact values of $p_i$ and $p_j$ can only be known after completion of jobs $i$ and $j$. At that point, the algorithm has also observed all relevant runtime signals, and it naturally has access to the problem inputs. Therefore, for practical use-cases of computable delays in non-clairvoyant algorithms (such as the application in Algorithm 2), the two definitions can be seen as equivalent.
>
> In the case of the algorithm following the predictions for progress bars with one signal (see Lemma 2.2 and the description in our initial rebuttal), $d(i,j)$ is a function of the three values $p_i$, $\beta_i p_i$, and $\beta_j p_j$. All of these quantities are known after both jobs $i$ and $j$ have completed:
> - $\beta_i p_i$ is the time at which job $i$ emits a signal, which the algorithm observes while processing it.
> - Similarly, $\beta_j p_j$ is observed during execution of job $j$.
> - $p_i$ is the total processing time of job $i$, revealed upon its completion.
>
> Thus, under the corrected definition, this algorithm satisfies the computable delays condition.
>
>
> Regarding the claim that most commonly used algorithms have computable delays, we clarify with the following examples:
> - Round-Robin, which is the optimal online algorithm, satisfies the condition, as $d(i,j) = \min(p_i, p_j)$, which is computable after both jobs are completed.
> - In the context of predicted processing times, SPJF (Shortest Predicted Job First) [PSK18] also satisfies the condition. In that setting: $d(i,j) = p_i \cdot \mathbf{1}(\pi_i \leq \pi_j)$, where $\pi_i$ and $\pi_j$ are the predicted processing times. Once job $i$ completes, $p_i$ is known, and $\pi_i$, $\pi_j$ are known from the input.
> - In the setting with permutation predictions [LM25a], where $\sigma$ is the predicted job order: $d(i,j) = p_i \cdot \mathbf{1}(\sigma(i) < \sigma(j))$, which depends on $p_i$, $\sigma(i)$, and $\sigma(j)$. Again, $p_i$ is revealed upon completion of job $i$, and $\sigma$ is available upfront.
>
> These three settings are the most common in scheduling with predictions. SPJF and following permutation predictions satisfy computable delays, and can therefore be combined with Round-Robin using our combining algorithm (rather than simple time-sharing), as established in Theorem 3.1 (see e.g. Section 3.3 for the application with permutation predictions).
>
>
> We thank the reviewer again for noting the incorrect definition (line 223). We will correct the definition of computable delays, and add more discussion in the main body of the paper to support our claim in Corollary 3.2. We would like to note that this modification does not affect the proofs of the paper.

---

### Official Review · Reviewer_FqMT · 2025-07-02

**Clarity:** 3
**Significance:** 3
**Originality:** 3
**Rating:** 4
**Confidence:** 3

**Summary:**

This paper introduces a novel model for non-clairvoyant scheduling enhanced with progress bars. The proposed framework interpolates between non-clairvoyant and clairvoyant settings by providing online feedback about job completion percentages, simulating real-world scenarios like system profiling or machine learning training. Three models of progress bars have been studied, including single signal progress bar, untrusted progress bar, stochastic progress bar. Besides, this paper combines worst-case competitive analysis with insights from learning-augmented algorithms and stochastic modeling. The authors design algorithms for each model and prove competitive ratio bounds. They also show how combining multiple scheduling strategies can improve performance and robustness.

**Questions:**

* Can the algorithms be applied to job scheduling scenarios with online, time-varying job arrivals?

**Ethical Concerns:**

["NO or VERY MINOR ethics concerns only"]

**Final Justification:**

I will keep my score as 4: borderline accept.

**Limitations:**

This paper does not have any negative societal impact.

**Paper Formatting Concerns:**

This paper does not have paper formatting concerns.

**Quality:**

3

**Strengths And Weaknesses:**

**Strengths.**
1. This paper presents a novel and natural model (scheduling with progress bars) for incorporating dynamic feedback into scheduling, which seems reasonable in some real-world scenarios. This broadens the algorithmic design space for the learning-augmented job scheduling problem.
2. The assumptions are well-justified and analyzed in both stochastic and adversarial settings.
3. This paper achieves an improved tradeoff (over time-sharing) compared with the SOTA learning-augmented scheduling algorithms.
4. The theoretical analysis is well-founded and clearly structured, which I appreciate.
5. The algorithm *Repeated Explore-Then-Commit* is intuitive and well motivated.

**Weaknesses.**
1. The new algorithm design is motivated by real-world phenomena (e.g., profiling and ML training in which multi-round progressive information can be obtained), but not evaluated using real-world job traces.
2. The empirical performance of the combining algorithm appears to be unstable and only shows clear advantages when $n$ is large. This limits its applicability, as it is uncommon for a large number of jobs to be queued on a single machine, particularly when such jobs, like those with progress bars, are typically ML training jobs discussed in this paper.

---

> ### Author Rebuttal · Authors · 2025-07-30
>
> We thank the reviewer for their interest in our work and for reading and assessing our paper carefully. In the following, we comment on weaknesses and questions raised by the reviewer.
>
> **Weakness:**
> Empirical performance is only advantageous for large $n$. We agree that our empirical experiments for the combining algorithm for $n=50$ (Figure 5) are not very stable (i.e., high variance). However, we emphasize that the average performance is still slightly better than previous methods, which are also unstable for a small number of jobs.
> Furthermore, we remark that for a very small number of jobs, the empirical and theoretical performance of a non-clairvoyant algorithm (without predictions) is already much better; e.g., the competitive ratio of Round-Robin is $4/3$ for $n=2$.
>
> **Question:**
> Algorithms for online job arrival. All of our algorithms can also be adapted to handle online job arrival and to minimize the sum of completion times. This is because our algorithms are based on local time-independent decisions. In the analysis, one then typically needs to account for release dates by charging the sum of release dates against an additional OPT, which usually results in an increase of the competitive ratio by at most 1. This is a classical technique for both clairvoyant and non-clairvoyant algorithms, see e.g., Schulz & Skutella “The power of α-points in preemptive single machine scheduling”, 2002, or Lindermayr & Megow, “Permutation Predictions for Non-Clairvoyant Scheduling”, 2025. We see a potential for relevant future work for online job arrival when minimizing the total flow time (that is, completion time minus release time). We will comment on these facts in the camera-ready version.

---

> > ### Comment · Reviewer_FqMT · 2025-08-04
> >
> > Thanks for the response. Although the authors does not address my concerns directly on empirical evaluation on real-world job traces, I would like to keep my positive score.

---

### Official Review · Reviewer_wQQL · 2025-07-03

**Clarity:** 3
**Significance:** 3
**Originality:** 3
**Rating:** 5
**Confidence:** 2

**Summary:**

This paper considers online scheduling problems where job sizes are unknown a priori but estimations about their remaining processing times are gradually revealed and refined. Online non-clairvoyant scheduling has been extensively studied in the literature. Recently, there has been a flurry of activities in online scheduling with predictions. Unlike these works, however, this paper studies the setup in which the scheduler starts with limited information although it is possible to get a better understanding via the means of progress bars. The paper considers three different progress bar models, for which efficient competitive algorithms are designed.

**Questions:**

- Can the framework of progress bars incorporate recent online machine learning schemes, such as multi-armed bandits, scheduling with testing, and queuing bandits?
- Progress bars allow more flexibility in comparison with scheduling with predictions. Is it possible to compare them in a more formal way?

**Ethical Concerns:**

["NO or VERY MINOR ethics concerns only"]

**Final Justification:**

I had questions regarding the connection to the existing work on scheduling with prediction and learning systems. The authors have addressed my questions. In accordance with this, I raise the score.

**Limitations:**

Yes

**Paper Formatting Concerns:**

No formatting concern.

**Quality:**

3

**Strengths And Weaknesses:**

Strengths
- This paper introduces a novel framework for online scheduling via the idea of progress bars. The framework is closely aligned with modern machine learning systems, where the decision-maker can obtain more refined information by analyzing more data sets. The paper proposes three models of progress bars, for which efficient algorithms have been devised.
- The paper provides strong theoretical guarantees in terms of strong competitive ratios that depend on the parameters of progress bar models.
- Numerical experiments demonstrate the computational performance of the algorithms.

Weaknesses
- As opposed to non-clairvoyant scheduling with predictions, the framework of this paper seems qualitatively different. However, formal comparisons are missing.

---

> ### Author Rebuttal · Authors · 2025-07-30
>
> We thank the reviewers for their interest and careful reading. As observed by the reviewer, the paper introduces and studies three models of progress bars, specifically:
> 1. Single-signal adversarial progress bar (Section 2)
> 2. Multiple-signal adversarial progress bars (Section 3)
> 3. Multiple-signal stochastic progress bars (Section 4)
>
> The adversarial models (1) and (2) have strong connections to *learning-augmented algorithms* (question 2), whereas the stochastic model (3) is more closely aligned with the *online learning* literature (question 1). We provide some further details on these connections in response to the reviewer’s questions.
>
> **Comparing the setting to algorithms with predictions (Sections 2 and 3).** Sections 2 and 3 of the paper are initially motivated by the *algorithms with predictions* literature (see lines 28-46). This literature has, for now, investigated scheduling problems under two main types of predictions:
> - *permutation predictions* (e.g. [PSK18]), where the permutation is an estimate of the ordering of the jobs’ processing times,
> - *processing time predictions* (e.g. [LM25a]), where the predictions are an estimate of the processing times’ values.
>
> These predictions are always provided to the scheduler along with the problem instance before the scheduling begins. Our progress bar models offer a strictly richer framework, where predictions typically refine over time and depend on the scheduler’s actions (see lines 63-66). We describe in the paper formal connections with classical models. The connection with processing time predictions is discussed in Section 2.3 (lines 196-208), where we rederive (and slightly improve) previous results. The connection with permutation predictions is discussed in Section 3.3 (lines 253-268), where we derive an algorithm improving upon previous results in a specific range of processing times.
>
> **Relation to bandits and bandits (Section 4).**  While there does not seem to be an immediate reduction between our setting and the bandit settings evoked by the reviewer, the problems appear to be closely connected. In particular, the scheduler relying on stochastic progress bars faces an explore-exploit tradeoff where it is incentivized to prioritize jobs for which the progress bar seems to progress faster (exploit) but must also allocate some computation to other jobs to refine estimates of their processing times (explore) (see lines 67-70 and 294-306). One crucial difference, however, is that we aim for a competitive ratio (i.e., multiplicative guarantee) rather than a regret (i.e., additive guarantee), in line with the literature on scheduling and online algorithms. Another (more superficial) difference is that our setting is in continuous time, whereas time in bandit problems is usually discrete. We note that the proof techniques in Section 4 draw inspiration from those in the bandits literature. Specifically, the lower bound (Theorem 4.3, proved in Section C.2) uses the fact that two progress bars are informationally indistinguishable (for a sufficiently long time) if their processing times (and thus, the rate of their reward) are sufficiently close. This argument is similar to those of [LS20, Chap. 15] or [BC 12, Sec 2.3] giving minimax lower bounds for stochastic bandits. The problems of scheduling with testing and queuing bandits will also be mentioned as related work.

---

> ### Comment · Reviewer_wQQL · 2025-08-03
>
> I appreciate the detailed exposition of connections between the framework of this paper and the existing works on scheduling with predictions, as well as online learning-based scheduling. I have noticed that the paper has already discussed comparisons with the previous works. Accordingly, I will raise my score.

---

### Decision · Program_Chairs · 2025-09-17

**Decision:**

Accept (spotlight)

**Comment:**

This paper proposes a new model for non-clairvoyant scheduling where the algorithm has access to progress bars that provide estimates of the fraction of each job that has been completed. Algorithms for different types of progress bars that achieve competitive ratio guarantees in terms of consistency and robustness are provided.
The proposed models with progress bars is novel, natural, and relevant. Different types of progress bars are explored. The proposed algorithms achieve strong and non-trivial theoretical guarantees.